# Set Diffusion: Interpolating Token Orderings Between Autoregression and Diffusion for Fast and Flexible Decoding

Marianne Arriola [1]    Volodymyr Kuleshov [1]

## Abstract

Discrete diffusion models have steadily improved in quality relative to autoregressive (AR) models. However, these models are normally constrained to fixed-length generation and do not support key-value (KV) caching. Block diffusion partially bridges diffusion and AR by generating token blocks left-to-right, but its fixed-size sequential blocks limit decoding flexibility and parallelism. Here, we present a new class of language models, set diffusion, comprised of (i) a likelihood parameterization that factorizes over flexible-position, flexible-length token sets and (ii) a set-causal diffusion architecture that supports KV cache updates after every inference step. By factorizing over token sets instead of fixed-size blocks, tokens can be decoded in arbitrarily-ordered sets, including sliding-window sets, enabling faster inference and support for any-order decoding. Set diffusion achieves better speed-quality tradeoffs on mathematical reasoning, summarization, and unconditional generation compared to prior diffusion language models while offering stronger infilling performance than block diffusion. We provide the code[1], along with the model weights and blog post on the project page: https://m-arriola.com/setdlms/

## 1. Introduction

Discrete diffusion models have recently gained traction in language modeling (Austin et al., 2021; Lou et al., 2023; Sahoo et al., 2024), music synthesis (Sun et al., 2022), and biological sequence design (Schiff et al., 2025). Compared to autoregressive (AR) models, they offer faster generation (Wu et al., 2025; Zheng et al., 2024) and improved controllability (Schiff et al., 2025; Li et al., 2022). However, they typically rely on architectures that preclude flexible-length generation (Kim et al., 2025a) and require full bidirectional context, which prevents key-value (KV) caching (Ma et al., 2025; Wu et al., 2025; Nguyen-Tri et al., 2025).

Block diffusion (Arriola et al., 2025a) partially mitigates these challenges by modeling sequences autoregressively over contiguous left-to-right blocks with discrete diffusion applied within each block. This yields a tighter likelihood bound, variable-length generation, and KV caching for previously decoded blocks. However, rigid left-to-right block structure restricts decoding flexibility for tasks requiring any-order generation such as infilling and arbitrary-position insertions (Li et al., 2025; Ma et al., 2026; Shu et al., 2026). Arriola et al. (2025a) also must delay KV cache updates until after a block is finished decoding because within-block denoising relies on bidirectional context (Wu et al., 2025; Seo et al., 2025). Finally, block diffusion limits parallel sampling since each block must be decoded before advancing (Liu et al., 2025b;a).

Our key observation is that diffusion language models need not interpolate between autoregression and diffusion by varying a fixed block size; instead, they can interpolate by varying the order in which token sets are generated. Based on this, we introduce set diffusion, which defines an autoregressive probability distribution over flexible-position, flexible-length sets of discrete random variables. Set diffusion generalizes block diffusion: fixed left-to-right blocks arise when sets are constrained to contiguous fixed-size blocks, while arbitrary-position and variable-length sets enable flexible insertion, infilling, and sliding-window decoding. By interpolating between left-to-right and fully random orderings, set diffusion improves the tradeoff between sample quality, inference efficiency, and decoding flexibility while supporting KV cache updates after every generation step. Across mathematical reasoning, summarization, and unconditional generation, set diffusion achieves better speed-quality tradeoffs than prior diffusion language models and substantially outperforms block diffusion on infilling.

---

[1]Department of Computer Science, Cornell University, Ithaca, NY, USA. Correspondence to: Marianne Arriola <marriola@cs.cornell.edu>.

*Proceedings of the 43rd International Conference on Machine Learning*, Seoul, South Korea. PMLR 306, 2026. Copyright 2026 by the author(s).

[1]Code: https://github.com/kuleshov-group/setdlms

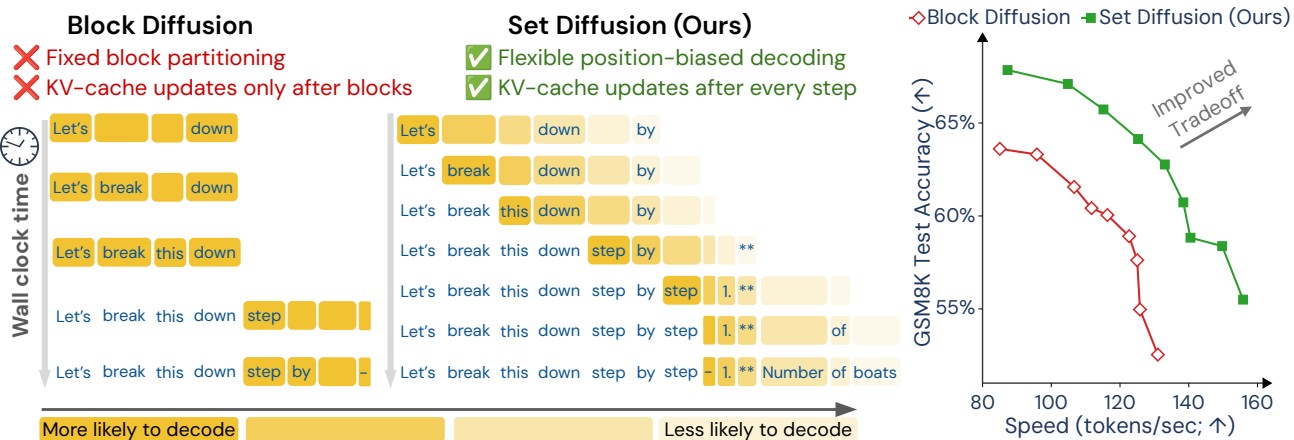

*Figure 1. Left:* Set diffusion generates tokens in arbitrary-position, arbitrary-length sets, biasing toward left-to-right decoding and updating the KV cache after each step. Block diffusion (Arriola et al., 2025a) is restricted to generate fixed-size sequential blocks and may only update the cache after each block completes. *Right:* Speed-accuracy tradeoffs on the GSM8K test (experimental details in Section L).

Our contributions are as follows:

1. We present set diffusion language models, which are autoregressive over flexible-length, flexible-position token sets, enabling variable-length generation and tunable generation orderings.

2. We introduce a set-causal diffusion architecture with KV cache updates after every inference step.

3. We show that set diffusion interpolates between AR and diffusion generation, enabling greater decoding flexibility and parallelism than block diffusion.

4. We demonstrate that set diffusion achieves state-of-the-art speed-quality tradeoffs among diffusion models across mathematical reasoning, infilling, summarization, and unconditional generation.

## 2. Background

**Notation** We denote scalar discrete random variables with $V$ categories as one-hot column vectors and define $\mathcal{V} = \{\mathbf{z} \in \{0,1\}^V : \sum_\ell \mathbf{z}_\ell = 1\} \subset \Delta^V$ to be the set of all such vectors, a subset of the simplex $\Delta^V$. We assume that the $V$-th category corresponds to a special [MASK] token and denote its one-hot vector by $\mathbf{m} \in \mathcal{V}$. We define $\mathbf{z}^{1:L}$ as a sequence of $L$ tokens, where $\mathbf{z}^\ell \in \mathcal{V}$, for $\ell \in 1, \ldots, L$, and let $\mathcal{V}^L$ denote the set of all such sequences. We write $\text{Cat}(\cdot; p)$ for a categorical distribution with parameters $p \in \Delta^V$, and $\odot$ for the Hadamard product.

### 2.1. Discrete Diffusion Models

Diffusion models (Sohl-Dickstein et al., 2015; Song & Ermon, 2019; Ho et al., 2020) are generative models that train

a denoising network $p_\theta$ to remove noise from latent variables $\mathbf{z}_t$ obtained from a corruption process $q$ that adds increasing noise to clean data $\mathbf{x}$ as $t$ increases. Discrete denoising diffusion probabilistic models (D3PM; Austin et al. (2021)) define corruption processes over discrete $\mathbf{z}_t$. MDLM (Sahoo et al., 2024) specializes to masked diffusion, which interpolates between clean data and a prior selected as $\mathbf{m}$: $q(\mathbf{z}_t|\mathbf{x}) = \text{Cat}(\mathbf{z}_t; \alpha_t \mathbf{x} + (1 - \alpha_t)\mathbf{m})$, where $\alpha_t = \alpha(t)$ is a noise schedule decreasing in $t$. The optimal diffusion model $p_\theta$ is the reverse of the forward process $q$. Sahoo et al. (2024) define $p_\theta(\mathbf{z}_s|\mathbf{z}_t) = q(\mathbf{z}_s|\mathbf{z}_t, \mathbf{x}_\theta(\mathbf{z}_t))$ where the denoising model $\mathbf{x}_\theta(\mathbf{x} \mid \mathbf{z}_t)$ predicts $\mathbf{x}$ given masked $\mathbf{z}_t$.

The diffusion model $p_\theta$ is fit by variational inference. Let $\text{KL}[\cdot]$ denote the Kullback-Leibler divergence. The Negative ELBO (NELBO) is given by (Sohl-Dickstein et al., 2015):

$$\mathcal{L}(\mathbf{x}; \theta) = \mathbb{E}_q \Big[ -\log p_\theta(\mathbf{x} \mid \mathbf{x}_{t(1)})$$
$$+ \sum_{j=1}^T D_{\text{KL}}\big(q(\mathbf{x}_{s(j)} \mid \mathbf{x}_{t(j)}, \mathbf{x}) \,\big\|\, p_\theta(\mathbf{x}_{s(j)} \mid \mathbf{x}_{t(j)})\big)$$
$$+ D_{\text{KL}}\big(q(\mathbf{x}_{t(T)} \mid \mathbf{x}) \,\big\|\, p_\theta(\mathbf{x}_{t(T)})\big) \Big] \tag{1}$$

This objective has natural continuous-time extensions based on continuous-time Markov chains (CTMCs), as well as score-based variants (Song & Ermon, 2019; Lou et al., 2023; Sun et al., 2022). Subsequent simplifications (Sahoo et al., 2024; Shi et al., 2024; Ou et al., 2024) tighten the NELBO and improve empirical performance.

### 2.2. Block Diffusion Language Modeling

Block diffusion language models (BD3LM; Arriola et al. (2025a)) interpolate between AR and order-agnostic diffu-

sion by factorizing the likelihood over $B$ blocks of length $S$. Adopting the shorthand from Arriola et al. (2025a), we denote tokens in block index $b \in [1, B]$ as $\mathbf{x}^b$ and preceding tokens as $\mathbf{x}^{<b}$. The likelihood factorizes as $\log p_\theta(\mathbf{x}) = \sum_{b=1}^{B} \log p_\theta(\mathbf{x}^b \mid \mathbf{x}^{<b})$, where each $p_\theta(\mathbf{x}^b \mid \mathbf{x}^{<b})$ is modeled using diffusion. A block-causal transformer parameterizes the reverse process $p_\theta(\mathbf{z}_s^b \mid \mathbf{z}_t^b, \mathbf{x}^{<b})$, permitting KV caching for each block once decoded. However, rigid blockwise factorization limits both flexibility and speed: each block must be decoded before the next, restricting parallel sampling and arbitrary-position editing (e.g., infilling), and KV caches may only be updated once a block is completed.

## 3. Set Diffusion

We introduce set diffusion models, which factorize generation over flexible-position, flexible-length token sets. By varying the token-set partition and ordering distribution, set diffusion recovers autoregression, fully non-autoregressive diffusion (Sahoo et al., 2024; Ou et al., 2024; Shi et al., 2024), and block diffusion (Arriola et al., 2025a) as special cases. We present a likelihood parameterization that supports variable-length sampling and a set-causal architecture that enables KV cache updates after every inference step.

### 3.1. A Probabilistic Model for Set Diffusion

Set diffusion combines autoregressive and diffusion language modeling by factorizing generation over arbitrary-position, arbitrary-length token sets and modeling each token-set conditional with diffusion.

**Definition 3.1** (Token set). A *position set* is a nonempty subset $\sigma_n \subseteq \{1, \ldots, L\}$ of token positions generated using diffusion. A position set sequence (or generation order) is a tuple $\sigma = (\sigma_1, \ldots, \sigma_N)$ of pairwise disjoint subsets over all positions where $\cup_{n=1}^{N} \sigma_n = \{1, \ldots, L\}$ and $\sigma_i \cap \sigma_j = \emptyset$ for $i \neq j$. A *token set* is the set of tokens corresponding to a position set, denoted $\mathbf{x}^{\sigma_n}$.

We marginalize over generation orders $\sigma$, each of which induces an autoregressive factorization over token sets $\mathbf{x}^{\sigma_1}, \ldots, \mathbf{x}^{\sigma_N}$. The resulting likelihood decomposes into position-selection probabilities from an order policy $\pi$ and conditional token-set likelihoods, following the decomposition from Turok et al. (2026):

$$p_\theta(\mathbf{x}) = \sum_\sigma \prod_{n=1}^{N} \pi(\sigma_n \mid \mathbf{x}^{\sigma_{<n}}) \cdot p_\theta(\mathbf{x}^{\sigma_n} \mid \mathbf{x}^{\sigma_{<n}}) \quad (2)$$

At step $n$, $\pi(\sigma_n \mid \mathbf{x}^{\sigma_{<n}})$ selects the next positions to generate and $p_\theta(\mathbf{x}^{\sigma_n} \mid \mathbf{x}^{\sigma_{<n}})$ assigns likelihood to the corresponding token values, where $\mathbf{x}^{\sigma_{<n}}$ denotes tokens at positions from the previously selected sets.

Directly optimizing (2) requires summing over combinatori-

ally many generation orders. We therefore apply Jensen's inequality over the latent order $\sigma$ with a variational distribution $q(\sigma \mid \mathbf{x})$. In this work, we assume a data- and parameter-independent order policy and choose $q(\sigma) \equiv \pi(\sigma)$ (Suppl. A).

We model each $p_\theta(\mathbf{x}^{\sigma_n} \mid \mathbf{x}^{\sigma_{<n}})$ using diffusion over a token set by introducing a reverse diffusion process as in Section 2.1, but restricted to set $n$: $p_\theta(\mathbf{z}_s^{\sigma_n} \mid \mathbf{z}_t^{\sigma_n}, \mathbf{x}^{\sigma_{<n}}) = q(\mathbf{z}_s^{\sigma_n} \mid \mathbf{z}_t^{\sigma_n}, \mathbf{x}_\theta(\mathbf{z}_t^{\sigma_n}, \mathbf{x}^{\sigma_{<n}}))$ where the denoising model $\mathbf{x}_\theta(\mathbf{z}_t^{\sigma_n}, \mathbf{x}^{\sigma_{<n}})$ predicts $\mathbf{x}^{\sigma_n}$ given noised $\mathbf{z}_t^{\sigma_n}$ and preceding clean sets in the generation order $\mathbf{x}^{\sigma_{<n}}$. Applying the NELBO in (1) to each conditional term in (2) yields the set diffusion objective $\mathcal{L}_{\text{SD}}(\mathbf{x}; \theta)$, which is itself a valid NELBO. Since the denoising model is conditioned on $\mathbf{x}^{\sigma_{<n}}$, we make this dependence on $\mathbf{x}^{\sigma_{<n}}, \theta$ explicit in $\mathcal{L}$:

$$-\log p_\theta(\mathbf{x}) \leq \mathcal{L}_{\text{SD}}(\mathbf{x}; \theta) := \mathbb{E}_{\sigma \sim \pi} \sum_{n=1}^{N} \mathcal{L}(\mathbf{x}^{\sigma_n}, \mathbf{x}^{\sigma_{<n}}; \theta)$$
$$(3)$$

In Section 5.1, we identify examples where training is tractable. In practice, $\pi$ can differ across training and inference: at inference, we can use sampling methods such as confidence-based heuristics (Wu et al., 2025).

The factorization in (2) supports the following capabilities:

1. **Variable-Length Generation.** At inference, we iteratively sample a position set $\sigma_n \sim \pi(\sigma_n \mid \mathbf{x}^{\sigma_{<n}})$ from the remaining ungenerated positions, then the corresponding token set $\mathbf{x}^{\sigma_n} \sim p_\theta(\mathbf{x}^{\sigma_n} \mid \mathbf{x}^{\sigma_{<n}})$.

2. **KV caching.** After generating a token set $\mathbf{x}^{\sigma_n}$, its key and value vectors can be cached and reused to condition subsequent predictions. We provide further details in Section 3.2.

3. **AR-Diffusion Interpolation.** Set diffusion flexibly tunes (i) parallelism, through the token-set size $|\sigma_n|$ and (ii) ordering bias, through the ordering distribution $\pi(\sigma_n \mid \sigma_{<n})$. Our framework generalizes several standard generative modeling paradigms:

   - **Autoregression** is recovered when $|\sigma_n| = 1$ with left-to-right orders (Suppl. F).
   - **Order-agnostic diffusion** (Sahoo et al., 2024; Shi et al., 2024; Ou et al., 2024) is recovered when $\pi$ is uniform (Suppl. B).
   - **Block diffusion** (Arriola et al., 2025a) is recovered when $\pi$ selects contiguous fixed-size blocks in left-to-right order (Suppl. C).
   - **Any-order AR** (Uria et al., 2014) is recovered when $|\sigma_n| = 1$ and $\pi$ is uniform (Suppl. D).
   - **FlexMDM** (Kim et al., 2025a) implements a learned $\pi$ where $\sigma_n$ is predicted by an auxiliary transformer output layer.

---

**Algorithm 1** Set Diffusion Training

---

**Require:** data distribution $q(\mathbf{x}^{1:L})$, order distribution $\pi(\sigma)$, model $\mathbf{x}_\theta$, forward process $q_t(\cdot|\mathbf{x})$, mask construction $\mathcal{M}$
  **repeat**
    $\mathbf{x}^{1:L} \sim q(\mathbf{x}^{1:L})$ {Sample datapoint}
    Sample $\sigma = (\sigma_1, \ldots, \sigma_N) \sim \pi(\sigma)$ {Set ordering}
    $\mathcal{M}_\sigma \leftarrow \mathcal{M}(\sigma)$ {Construct set-causal mask}
    Sample $t_1, \ldots, t_N \sim \mathcal{U}[0,1]$
    $\forall n \in \{1, \ldots, N\}: \mathbf{z}_{t_n}^{\sigma_n} \sim q_{t_n}(\cdot \mid \mathbf{x}^{\sigma_n})$
    $\mathbf{x}_{\text{in}} \leftarrow \mathbf{x}^{\sigma_{1:N}} \oplus \mathbf{z}_{t_{1:N}}^{\sigma_{1:N}}$ {Construct denoiser input}
    $\mathbf{x}_{\text{logit}}^{\sigma_{1:N}} = \mathbf{x}_\theta(\mathbf{x}_{\text{in}}, \mathcal{M}_\sigma)$
    Take gradient step on $\nabla_\theta \mathcal{L}_{\text{SD}}(\mathbf{x}_{\text{logit}}^{\sigma_{1:N}}; \theta)$
  **until** converged

---

**Algorithm 2** Parallel Set Diffusion Sampling

---

**Require:** Prompt $\mathbf{x}^{1:P}$, model $\mathbf{x}_\theta$, diffusion sampler SAMPLE, # steps $N$
  $\emptyset, \mathbf{K}, \mathbf{V} \leftarrow \mathbf{x}_\theta(\mathbf{x}^{1:P}, \emptyset)$ {Cache prompt}
  $\sigma_0 \leftarrow \{1, \ldots, P\}$
  **for** $n = 1$ to $N$ **do**
    $\tilde{\sigma}_n \sim \pi(\tilde{\sigma}_n | \sigma^{<n})$ {Select candidate positions}
    $\sigma_n, \mathbf{x}^{\sigma_n} \leftarrow \text{SAMPLE}(\mathbf{x}_\theta^{\tilde{\sigma}_n}, \mathbf{K}^{\sigma^{<n}}, \mathbf{V}^{\sigma^{<n}})$
    $\mathbf{x} \leftarrow \mathbf{x}^{\sigma^{<n}} \oplus \mathbf{x}^{\sigma_n}$ {Keep accepted tokens}
    $\emptyset, \mathbf{K}^{\sigma_n}, \mathbf{V}^{\sigma_n} \leftarrow \mathbf{x}_\theta^{\sigma_n}(\mathbf{x}^{\sigma_n}, \mathbf{K}^{\sigma^{<n}}, \mathbf{V}^{\sigma^{<n}})$
    $(\mathbf{K}, \mathbf{V}) \leftarrow (\mathbf{K}^{\sigma^{<n}} \oplus \mathbf{K}^{\sigma_n}, \mathbf{V}^{\sigma^{<n}} \oplus \mathbf{V}^{\sigma_n})$
    $\sigma \leftarrow \sigma_{<n} \cup \sigma_n$
  **end for**
  **return** $\mathbf{x}^\sigma$

---

## 3.2. Set Diffusion Architecture

We parameterize each term $p_\theta(\mathbf{x}^{\sigma_n} \mid \mathbf{x}^{\sigma_{<n}})$ using a single neural network $\mathbf{x}_\theta$, following the same notation from Arriola et al. (2025a). Specifically, we parameterize $\mathbf{x}_\theta$ using a transformer (Vaswani et al., 2017) which outputs token probabilities and key-value caches $\mathbf{K}^{\sigma_n}, \mathbf{V}^{\sigma_n}$. Unlike block diffusion, where the denoiser predicts a fixed contiguous block, set diffusion may query an arbitrary set of positions depending on the ordering distribution $\pi(\sigma)$. When $\mathbf{x}_\theta$ is trained, $\mathbf{x}_\theta^{\sigma_n}(\mathbf{z}_t^{\sigma_n}, \mathbf{x}^{\sigma_{<n}})$ yields predictions for denoised tokens in set $\sigma_n$ based on noised $\mathbf{z}_t^{\sigma_n}$ and clean $\mathbf{x}^{\sigma_{<n}}$. The transformer uses a set-causal attention mask, where tokens in a set $\sigma_n$ attend to tokens in sets $\sigma_1, \ldots, \sigma_n$ (Suppl. K).

As in autoregressive decoding, previously generated token sets need not be reprocessed at every step; their keys and values can instead be cached and reused. We denote the keys and values for set $\sigma_n$ by $\mathbf{K}^{\sigma_n}$ and $\mathbf{V}^{\sigma_n}$, and define $\mathbf{x}_\theta$ to both consume cached context and return updated cache entries. Its full signature is

$$\mathbf{x}_{\text{logit}}^{\sigma_n}, \mathbf{K}^{\sigma_n}, \mathbf{V}^{\sigma_n} \leftarrow \mathbf{x}_\theta^{\sigma_n}(\mathbf{z}_t^{\sigma_n}, \mathbf{K}^{\sigma^{<n}}, \mathbf{V}^{\sigma^{<n}})$$
$$:= \mathbf{x}_\theta^{\sigma_n}(\mathbf{z}_t^{\sigma_n}, \mathbf{x}^{\sigma_{<n}}) \quad (4)$$

where $\mathbf{x}_{\text{logit}}^{\sigma_n}$ denotes the predictions for clean $\mathbf{x}^{\sigma_n}$ and $\mathbf{K}^{\sigma_{<n}}, \mathbf{V}^{\sigma_{<n}}$ are the keys and values cached on a forward pass of $\mathbf{x}_\theta$ over $\mathbf{x}^{\sigma_{<n}}$ (hence the inputs $\mathbf{x}^{\sigma_{<n}}$ and $\mathbf{K}^{\sigma_{<n}}, \mathbf{V}^{\sigma_{<n}}$ are equivalent).

## 3.3. Training

**Objective** We train set diffusion models using maximum likelihood estimation by minimizing the NELBO in (3). Training is especially tractable for a fixed ordering distribution $\pi(\sigma)$ that is independent of the data and model parameters. In this setting, we may first sample $\sigma \sim \pi$ and then approximate $\mathcal{L}_{\text{SD}}(\mathbf{x}; \theta)$ using Monte Carlo with a single ordering $\sum_{n=1}^N \mathcal{L}(\mathbf{x}^{\sigma_n}, \mathbf{x}^{\sigma_{<n}}; \theta)$. Learned orderings

are also compatible with set diffusion, but optimizing them requires propagating gradients through discrete sampling (Wang et al., 2025c); we leave this for future work.

**Optimization** We train with mini-batch gradient descent using an efficient algorithm that computes likelihoods for all token sets in a single forward pass (Alg. 1). Ideally, one would use a single forward pass of $\mathbf{x}_\theta$ to estimate $p_\theta(\mathbf{z}_s^{\sigma_n}|\mathbf{z}_t^{\sigma_n}, \mathbf{x}^{\sigma_{<n}})$ for $n = 1, \ldots, N$. However, the denoising loss for set $\mathbf{x}^{\sigma_n}$ is computed from its noisy version $\mathbf{z}_t^{\sigma_n}$, whereas subsequent sets condition on the clean version $\mathbf{x}^{\sigma_n}$. Thus, each token set must appear both as clean context and as a noisy query.

We propose a vectorized implementation following Arriola et al. (2025a) to compute $\mathcal{L}_{\text{SD}}(\mathbf{x}; \theta)$ in one forward pass by concatenating the clean and noisy views:

$$\mathbf{x}_{\text{in}} = \underbrace{\mathbf{x}^{\sigma_1} \oplus \cdots \oplus \mathbf{x}^{\sigma_N}}_{\text{clean sets}} \oplus \underbrace{\mathbf{z}_{t_1}^{\sigma_1} \oplus \cdots \oplus \mathbf{z}_{t_N}^{\sigma_N}}_{\text{noisy query sets}} \quad (5)$$

where $t_n$ is the noise level applied to token set $\mathbf{x}^{\sigma_n}$. We design an ordering-dependent attention mask for $\mathbf{x}_{\text{in}}$ so that noisy tokens attend to other noisy tokens in their set and to all clean tokens in preceding sets (see Suppl. K).

## 3.4. Inference

At inference, we ancestrally sample token sets using Algorithm 2. At step $n$, we sample candidate positions $\tilde{\sigma}_n \sim \pi(\tilde{\sigma}_n \mid \mathbf{x}^{\sigma_{<n}})$ and call any sampling procedure

$$\sigma_n, \mathbf{x}^{\sigma_n} \leftarrow \text{SAMPLE}(\mathbf{x}_\theta^{\tilde{\sigma}_n}, \mathbf{K}^{\sigma^{<n}}, \mathbf{V}^{\sigma^{<n}}), \quad (6)$$

which samples from $p_\theta(\mathbf{x}^{\tilde{\sigma}_n} \mid \mathbf{x}^{\sigma_{<n}})$ and may accept $\sigma_n \subseteq \tilde{\sigma}_n$. Accepted tokens are committed and passed through the transformer once to append $\mathbf{K}^{\sigma_n}, \mathbf{V}^{\sigma_n}$ to the cache. Note that this procedure allows the use of a different ordering distribution $\pi$ at inference than at training, e.g., to implement confidence-based heuristics (Wu et al., 2025).

# 4. Interpolating Between Token Orderings

We now describe a structured family of ordering distributions $\pi(\sigma)$ that interpolates between order-agnostic diffusion and autoregression by changing the order in which token positions are revealed. We specialize to fixed, data- and parameter-independent ordering distributions induced by position-dependent reveal-time schedules. This interpolation controls the speed–quality tradeoff in set diffusion: order-agnostic generation enables larger, more flexible token sets for parallel decoding, while stronger left-to-right bias tightens the NELBO and improves sample quality.

Block diffusion (Arriola et al., 2025a) instantiates this tradeoff by varying a block size, but its fixed contiguous blocks restrict arbitrary-position insertion. Instead, we parameterize $\pi(\sigma)$ through position-dependent reveal-time distributions: each token position is assigned a random reveal time, and sorting the sampled reveal times induces a generation order. The resulting position-offset schedules provide a simple mechanism for controlling left-to-right bias while retaining flexible-position decoding.

## 4.1. Token Ordering via Position-Dependent Schedules

Rather than parameterizing $\pi(\sigma)$ directly over the combinatorial space of orderings, we induce it from position-dependent ordering schedules. Intuitively, each position $\ell$ is assigned a random reveal time $R^\ell \in [0, 1]$; positions whose reveal-time distributions place more mass near zero are more likely to appear earlier in the generation order.

For each position $\ell$, let $\alpha_\tau^\ell \in [0, 1]$ be a monotonically increasing schedule over ordering time $\tau \in [0, 1]$. The quantity $\alpha_\tau^\ell$ is the probability that position $\ell$ has been revealed by time $\tau$, with boundary conditions $\alpha_0^\ell = 0$ and $\alpha_1^\ell = 1$. Hence, $\Pr(R^\ell \leq \tau) = \alpha_\tau^\ell$ defines the CDF of the token-specific reveal time $R^\ell$, where $R^\ell$ has density $f_\ell(\tau) = \dot{\alpha}_\tau^\ell$.

We independently sample $R^\ell$ for each position and sort the sampled reveal times in ascending order to obtain a generation ordering $\sigma = \text{argsort}(R^1, \ldots, R^L)$. Reveal times can also be rounded to a grid with $m$ intervals on $[0, 1]$ as $\tilde{R}^\ell = \frac{1}{m} \arg\min_{i \in \{0,\ldots,m\}} \left| \frac{i}{m} - R^\ell \right|$ to induce a position sequence of non-singleton sets, where each set groups together identical rounded reveal times.

This construction can be viewed as a multivariate masked diffusion schedule over token positions; in Suppl. E, we derive the corresponding NELBO and show that, unlike the univariate case, the objective depends on the trajectory of the token-wise schedule, following similar findings for Gaussian diffusion (Sahoo et al., 2023).

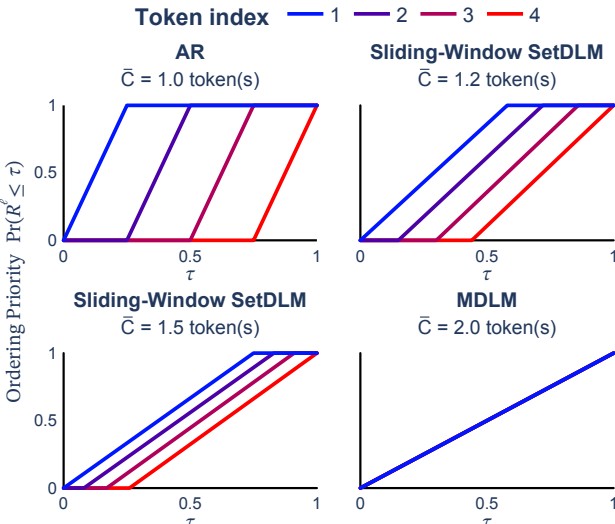

*Figure 2.* Position-offset reveal-time CDFs for $L = 4$ tokens. For the $\ell$-th token, $R^\ell \in [0, 1]$ is its reveal time and $\Pr(R^\ell \leq \tau)$ is the probability that token $\ell$ has been revealed by normalized ordering time $\tau \in [0, 1]$. The decoding width $w$ controls the ordering bias, interpolating between AR and order-agnostic diffusion generation. $\bar{C}$ denotes the expected inference prediction budget (Def. 4.2).

## 4.2. Position-Offset Schedules

We instantiate $\boldsymbol{\alpha}_\tau$ as evenly spaced monotone schedules that bias generation toward left-to-right orderings. Specifically, token $\ell \in \{1, \ldots, L\}$ has an active generation interval $[a^\ell, a^\ell + w]$ with window size $w \in (0, 1]$ (Figure 2). Before this interval, the token is selected with zero probability; after this interval, it is guaranteed to have been generated.

$$\alpha_\tau^\ell = \begin{cases} 0 & \text{if } \tau \leq a^\ell \\ \frac{\tau - a^\ell}{w} & \text{if } a^\ell < \tau < a^\ell + w \\ 1 & \text{if } \tau \geq a^\ell + w \end{cases} \quad (7)$$

The window size $w$ controls the strength of the left-to-right bias—smaller $w$ reduces overlap between active generation intervals, making earlier tokens likely to be generated first.

## 4.3. Properties

Position-dependent ordering schedules control the distribution of prediction tasks seen during training and inference. For position-offset schedules as $w \to \frac{1}{L}$, the ordering distribution concentrates on left-to-right generation orders. We establish the following result in Suppl. F.

**Proposition 4.1.** *As the unmasking width $w \to 1/L$, the per-token ELBO tends to an autoregressive objective. In the limit, this holds true with equality and the ELBO is tight.*

The generation window size $w$ interpolates between MDLM and AR. When $w = 1$, all positions share the same schedule, $\alpha_\tau^\ell = \tau$, recovering order-agnostic generation (e.g., MDLM

(Sahoo et al., 2024; Shi et al., 2024; Ou et al., 2024)); when $w = \frac{1}{L}$, the active generation intervals are non-overlapping and only one token may be generated at a time, recovering AR generation (Suppl. F). Thus, smaller $w$ improves perplexity, while larger $w$ permits more parallel and arbitrary-position generation. To compare set diffusion and block diffusion under matched inference parallelism, we use the "expected inference prediction budget" (Suppl. H):

**Definition 4.2** (Expected inference prediction budget)**.** The expected number of tokens eligible for prediction at a given ordering time, averaged over $\tau \in [0, 1]$. A token at position $\ell$ is eligible at ordering time $\tau$ if it has not yet been generated and $\tau$ lies in its active generation interval:

$$\bar{C} = \int_0^1 \sum_{\ell=1}^{L} \Pr(R^\ell > \tau) \cdot \mathbf{1}\{\dot{\alpha}_\tau^\ell \neq 0\} \, \mathrm{d}\tau \qquad (8)$$

# 5. Sliding-Window Set Diffusion Language Models (SW-SetDLMs)

We introduce Sliding-Window Set Diffusion Language Models (SW-SetDLMs), a practical instantiation of set diffusion with a position-offset ordering distribution $\pi(\sigma)$, a factorized token-set likelihood, and an order-causal transformer architecture. These choices enable variable-length sliding-window decoding, KV cache updates after each inference step, and low-variance training. Unlike block diffusion (Arriola et al., 2025a), SW-SetDLMs decode flexible-position, flexible-length sets while preserving a left-to-right bias for high-quality generation.

## 5.1. Probabilistic Model

SW-SetDLMs instantiate the set diffusion likelihood in (2) with two modeling choices. First, we use an order distribution $\pi(\sigma \mid \mathbf{x}) = \pi(\sigma)$ induced by a position-offset schedule whose mass is biased toward left-to-right orders. This bias controls interpolation between order-agnostic diffusion and autoregression: stronger left-to-right bias yields a tighter NELBO, while weaker bias permits increased parallelism.

Second, we use a factorized one-shot denoising likelihood within each token set, $p_\theta(\mathbf{x}^{\sigma_n} \mid \mathbf{x}^{<\sigma_n}) = \prod_{\ell \in \sigma_n} p_\theta(\mathbf{x}^\ell \mid \mathbf{x}^{<\sigma_n})$, similar to Turok et al. (2026). As a result, the set size $|\sigma_n|$ is the primary control on inference parallelism: all tokens in $\sigma_n$ are predicted simultaneously conditioned on the same revealed context $\mathbf{x}^{\sigma_{<n}}$, and the generated tokens can be committed to the KV cache before the next generation step. Equivalently, SW-SetDLMs use masked diffusion restricted to the selected set with a single denoising step, $T = 1$: the forward process fully masks a set and the reverse denoising model $\mathbf{x}_\theta(\mathbf{m}^{\sigma_n}, \mathbf{x}^{\sigma_{<n}})$ predicts each token $\mathbf{x}^\ell$ for $\ell \in \sigma_n$ in parallel from clean $\mathbf{x}^{\sigma_{<n}}$. This assumes conditional independence across tokens in the set $\{\mathbf{x}^\ell : \ell \in \sigma_n\}$ given the revealed context.

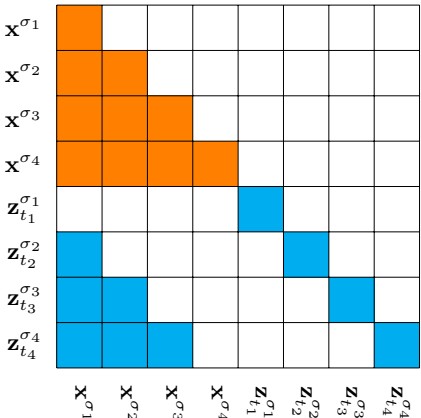

*Figure 3.* Causal attention mask for $L = 4$ singleton token sets, ordering $\sigma$, clean tokens $\mathbf{x}^{\sigma_{1:N}}$, and corrupted tokens $\mathbf{z}_{t_{1:N}}^{\sigma_{1:N}}$.

## 5.2. Architecture

SW-SetDLMs use a set-causal transformer whose attention pattern follows the sampled generation order. During training, singleton token sets allow each input sequence to be permuted into generation order, reducing set-causal attention to a reusable standard causal mask that avoids per-order mask construction and is compatible with compiled block-sparse attention implementations (Dong et al., 2024).

## 5.3. Training

**Objective** While set diffusion permits arbitrary token sets, SW-SetDLMs specialize training to singleton sets for architectural and training efficiency. Singleton sets allow a fixed causal mask (Section 5.2) and enable a token-efficient, low-variance estimator in which each sampled ordering contributes all $L$ conditional likelihoods. Concretely, we sample a full ordering over token positions and predict each token conditioned on tokens that precede it in that ordering:

$$-\log p_\theta(\mathbf{x}) \leq \mathcal{L}_{\mathrm{SD}}(\mathbf{x}; \theta) := -\mathbb{E}_{\sigma \sim \pi} \sum_{n=1}^{L} \log p_\theta(\mathbf{x}^{\sigma_n} | \mathbf{x}^{\sigma_{<n}})$$

(9)

We allow non-uniform $\pi$ (Suppl. A), unlike Hoogeboom et al. (2021), to emphasize favorable generation orderings and tighten the NELBO (Prop. 4.1). A masked-diffusion-style estimator follows by sampling an ordering step and averaging over possible next tokens (Suppl. B):

$$-\log p_\theta(\mathbf{x})$$
$$\leq -N \, \mathbb{E}_{n, \sigma_{<n}} \sum_{k \in \sigma_{\geq n}} \pi(\sigma_n = k \mid \sigma_{<n}) \log p_\theta(\mathbf{x}^k \mid \mathbf{x}^{\sigma_{<n}})$$

(10)

When $\pi$ is uniform, (10) recovers the standard masked diffusion objective (Sahoo et al., 2024; Shi et al., 2024; Ou et al., 2024). Under strong ordering bias, however, $\pi(\sigma_n = k \mid \sigma_{<n})$ concentrates on a few eligible positions, so (10) receives gradient signal from only a few token predictions per mini-batch (Arriola et al., 2025a). We optimize (9), which uses all $L$ conditional factors per sampled ordering and yields lower-variance training (see Table 5).

*Table 1.* Evaluation on GSM8K test set. Best diffusion value is **bolded**. We report decoding throughput (Tput) in tokens / sec over 200 generations on an H100 80GB GPU. We use $T = L$ total sampling steps, so the throughput can be higher for diffusion models when $T < L$. For MDLM (Sahoo et al., 2024), we use confidence-aware parallel decoding with threshold 0.9 and block-wise decoding with $S = 32$ for tractable inference. See Section L for experimental details.

|  | PPL ($\downarrow$) | 0-shot pass@1($\uparrow$) | Tput ($\uparrow$) |
|---|---|---|---|
| AR Transformer | 1.25 | 75.74 | $67.16_{\pm 0.84}$ |
| MDLM | $\leq 2.10$ | 6.37 | $\geq 24.48_{\pm 3.82}$ |
| BD3LM $S = 16$ | $\leq 1.67$ | 50.49 | $\geq 52.18_{\pm 0.48}$ |
| $S = 8$ | $\leq 1.53$ | 56.94 | $\geq 54.67_{\pm 0.61}$ |
| $S = 4$ | $\leq \mathbf{1.41}$ | 63.53 | $\geq 55.39_{\pm 0.62}$ |
| SW-SetDLM $S \leq 32$ | $\leq 1.59$ | 61.94 | $\geq 55.08_{\pm 1.09}$ |
| $S \leq 16$ | $\leq 1.50$ | 63.91 | $\geq 56.50_{\pm 0.75}$ |
| $S \leq 8$ | $\leq 1.42$ | **66.41** | $\geq \mathbf{60.42_{\pm 0.14}}$ |

**Optimization**  We optimize SW-SetDLMs with set diffusion training specialized to singleton token sets (Section 3.3). For each sampled ordering, a single causal forward pass computes all $L$ conditional terms in Eq. 9. Unlike prior masked diffusion objectives, which compute gradients only from masked tokens (Arriola et al., 2025a; Sahoo et al., 2024), SW-SetDLM uses every token position, reducing gradient variance. The same objective supports one-pass full-sequence likelihood computation, useful for applications including efficient policy-gradient estimation in reinforcement learning (Wang et al., 2025b; Sahoo et al., 2025b).

### 5.4. Inference

SW-SetDLM inference instantiates set diffusion sampling (Section 3.4) with the position-offset ordering schedule. At step $n$, the schedule selects candidate positions $\tilde{\sigma}_n$ from the ungenerated tokens whose active generation intervals $[a^\ell, a^\ell + w]$ contain the current ordering time $\tau_n$. We restrict $\tilde{\sigma}_n$ to a sliding output window of size at most $S_{\max}$ tokens and denoise all positions $\tilde{\sigma}_n$ in parallel from the current KV cache.

The sampler accepts $\sigma_n \subseteq \tilde{\sigma}_n$ and assigns token values $\mathbf{x}^{\sigma_n}$. Accepted tokens are committed, ordered according to the schedule-induced causal order, and passed through the transformer once to append $\mathbf{K}_{\sigma_n}, \mathbf{V}_{\sigma_n}$ to the cache.

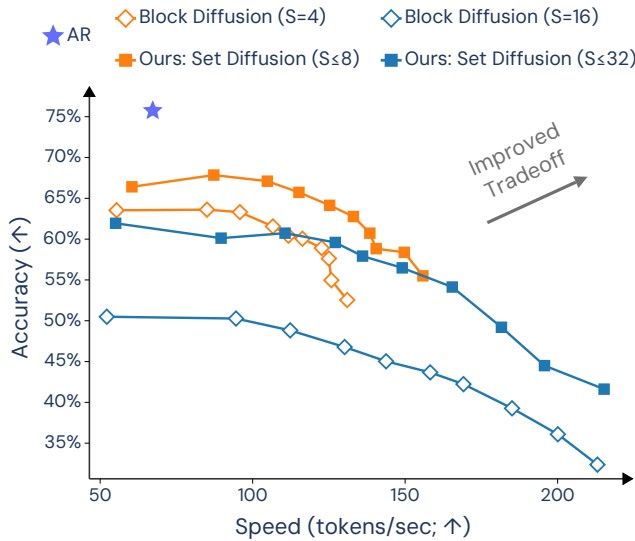

*Figure 4.* Set diffusion achieves better speed-accuracy tradeoffs on the GSM8K test set compared to block diffusion (Arriola et al., 2025a), where $S$ denotes the training output window size. We report decoding throughput (Tput) in tokens / sec on an H100 80GB GPU. Details in Section L.

## 6. Experiments

We evaluate set diffusion on mathematical reasoning, summarization, unconditional generation, and likelihood estimation (full details in Suppl. L). We train set diffusion models by varying the decoding window size $w$ to match the inference prediction budget (Def. 4.2) of block diffusion models of block sizes $S \in \{4, 8, 16\}$ (see Suppl. H). Unlike block diffusion, we tune our schedule to allow any-order decoding up to $2S$ positions ahead, providing greater decoding flexibility for tasks such as infilling. Thus, our set diffusion models have maximum window sizes $S_{\max} \in \{8, 16, 32\}$.

We examine 1) mathematical reasoning (GSM8K; Cobbe et al. (2021)) with zero-shot pass@1 accuracy, as well as 2) text summarization (CNN/DailyMail; Hermann et al. (2015); See et al. (2017)) and 3) infilling (Mostafazadeh et al., 2016) for which we compute ROUGE scores (Lin, 2004). We also train set diffusion models on pretraining datasets OpenWebText (OWT; Gokaslan et al. (2019)) and One Billion Words (LM1B; Chelba et al. (2014)), for which we compute perplexity (PPL) on their validation splits.

### 6.1. Mathematical Reasoning

On GSM8K, set diffusion shows improved accuracy and decoding throughput compared to diffusion baselines. In Figure 4, we map the speed-quality Pareto frontier, showing that set diffusion attains a strictly better tradeoff than block diffusion. We provide qualitative results in Suppl. N.2.

| Model | Size | Tokens | Infill 1/5 | | Infill 3/5 | |
|---|---|---|---|---|---|---|
| | | | *R-1 / 2 / L* (↑) | Tput (↑) | *R-1 / 2 / L* (↑) | Tput (↑) |
| *Autoregression* | | | | | | |
| GPT2-S[†] (Radford et al., 2019) | 127M | n/a | 9.5 / 0.4 / 8.7 | - | 13.5 / 0.6 / 10.2 | - |
| AR Transformer (Sahoo et al., 2024) | 130M | 157B | 8.2 / 0.5 / 7.6 | $159.7_{\pm 1.8}$ | 18.3 / 1.3 / 13.1 | $158.4_{\pm 0.8}$ |
| *Diffusion* | | | | | | |
| SEDD-S[†] (Lou et al., 2023) | 170M | 210B | 11.6 / 0.8 / 10.7 | - | 16.2 / 1.3 / 12.2 | - |
| MDLM (Sahoo et al., 2024) | 130M | 157B | 14.5 / 1.6 / 13.3 | $\geq 71.9_{\pm 2.9}$ | 22.2 / 2.3 / 15.2 | $\geq 72.6_{\pm 2.7}$ |
| DiffuGPT-S[†] (Gong et al., 2025) | 127M | n/a | 14.0 / 1.5 / 13.0 | - | 16.4 / 2.0 / 14.2 | - |
| ASSD[†] (Guo & Ermon, 2025) | 110M | 45B | 13.1 / 1.1 / 12.0 | - | 18.0 / 1.4 / 13.2 | - |
| *AR+Diffusion Hybrid* | | | | | | |
| BD3LM $S = 16$ (Arriola et al., 2025a) | 110M | 157B | 9.2 / 0.6 / 8.6 | $\geq 105.8_{\pm 4.5}$ | 15.8 / 0.8 / 11.1 | $\geq 114.2_{\pm 1.3}$ |
| SW-SetDLM $S \leq 32$ (Ours) | 110M | 157B | 11.6 / 1.0 / 10.9 | $\geq 132.0_{\pm 1.8}$ | 18.1 / 1.3 / 13.2 | $\geq 123.1_{\pm 2.6}$ |

*Table 2.* ROCStories infilling ROUGE scores (↑). "Infill $k/5$", infills $k$ middle sentences given the rest. We report decoding throughput (Tput) in tokens / sec on an H100 80GB GPU. Diffusion models use the maximum number of sampling steps, thus we report throughput as a lower bound. † indicates values reported from Guo & Ermon (2025).

## 6.2. Infilling

Following Gong et al. (2025), we evaluate infilling on 1871 five-sentence ROCStories examples (Mostafazadeh et al., 2016) using OWT-trained models. Set diffusion infills one or three middle sentences and achieves higher ROUGE than block diffusion with up to ∼25% faster decoding (Table 2). MDLM is substantially slower because it lacks KV caching and recomputes the full 1024-token context at every denoising step, despite ROCStories examples being much shorter. Qualitative samples are in Suppl. N.1.

## 6.3. Summarization

On text summarization, set diffusion achieves competitive ROUGE scores compared to diffusion baselines and up to 10% faster decoding than block diffusion (Table 3). Set diffusion outperforms an AR baseline, which we find overfits on CNN/DailyMail. In contrast, diffusion models train on diverse prediction tasks that mitigate overfitting (Prabhudesai et al., 2025).

## 6.4. Unconditional Sampling and Likelihood Estimation

We evaluate sample quality on OWT with MAUVE (Pillutla et al., 2021). Table 4 shows that set diffusion improves MAUVE over block diffusion while decoding faster.

On LM1B, set diffusion achieves the best test perplexity among diffusion models in Table 9, particularly for window size $S \leq 8$, while maintaining consistently higher decoding throughput than prior diffusion models. On OWT (Suppl. M), when block diffusion and set diffusion have comparable likelihoods, set diffusion achieves 22% faster decoding.

*Table 3.* CNN/DailyMail test set ROUGE scores (↑). We report decoding throughput (Tput) over 1k samples in tokens / sec on an H100 80GB GPU. Diffusion models are evaluated with $T = L$ total sampling steps; fewer steps ($T < L$) can increase throughput.

| | | ROUGE (↑) | | |
|---|---|---|---|---|
| | | 1 | 2 | L |
| *Past baselines* | | | | |
| GPT-2 (Radford et al., 2019) | | 29.3 | 8.3 | 26.6 |
| BERT-L (Liu & Lapata, 2019) | | 41.7 | 19.4 | 38.8 |
| T5-L (Raffel et al., 2020) | | 42.5 | 20.7 | 39.8 |
| AR-Diff. ($k = 50$) (Wu et al., 2023a) | | 39.6 | 16.3 | 37.1 |
| GENIE ($k = 50$) (Lin et al., 2023) | | 29.3 | 8.3 | 21.9 |
| | Tput (↑) | 1 | 2 | L |
| AR Transformer | $52.4_{\pm 0.9}$ | 32.2 | 11.6 | 22.0 |
| MDLM | $\geq 32.3_{\pm 0.8}$ | 35.9 | 15.1 | 23.6 |
| BD3LM $S = 16$ | $\geq 38.8_{\pm 0.2}$ | 40.0 | 17.8 | 27.0 |
| $S = 8$ | $\geq 38.9_{\pm 0.3}$ | 39.8 | 17.4 | 26.8 |
| $S = 4$ | $\geq 41.3_{\pm 0.4}$ | 38.8 | 16.7 | 26.4 |
| SW-SetDLM (Ours) $S \leq 32$ | $\geq 42.3_{\pm 0.8}$ | 38.1 | 17.3 | 27.3 |
| $S \leq 16$ | $\geq 42.8_{\pm 0.5}$ | 38.2 | 17.2 | 27.1 |
| $S \leq 8$ | $\geq 44.7_{\pm 0.8}$ | 38.0 | 16.8 | 26.7 |

## 6.5. Ablations

We ablate two set diffusion design choices under a fully order-agnostic schedule: (1) the causal architecture enabling efficient KV caching, and (2) token-efficient training. Table 6 shows that SW-SetDLMs achieve higher decoding throughput and improved GSM8K accuracy compared to a bidirectional variant without KV caching or token-efficient training. Table 5 shows that leveraging all tokens in a batch reduces gradient variance by ∼ 3× and accelerates convergence compared to naive training (Sahoo et al., 2024).

## 7. Related Work, Discussion, and Conclusion

**Comparison to MDLMs** Set diffusion builds on denoising discrete diffusion (Austin et al., 2021) and recent

*Table 4.* MAUVE (↑) of 1k samples of maximum length $L \leq 1024$. We report decoding throughput (`Tput`) in tokens / sec on 4 `A6000 48GB` GPUs. Diffusion models are evaluated with $T = L$ total sampling steps; using fewer steps ($T < L$) can increase throughput. Details in Suppl. L. **Bold**: best diffusion value.

| Model | MAUVE (↑) | Entropy | Tput (↑) |
|---|---|---|---|
| AR | 92.94 | 5.02 | $97.80_{\pm 1.27}$ |
| BD3LM $S = 16$ | 64.91 | 4.37 | $\geq 81.22_{\pm 0.68}$ |
| SW-SetDLM $S \leq 32$ | **71.98** | 4.23 | $\geq \mathbf{91.30_{\pm 0.71}}$ |

*Table 5.* LM1B test perplexity and gradient variance (Suppl. K.1) over the last 10 gradient updates from training over 50K examples (3.3B tokens). **Bold**: best diffusion value.

| | PPL (↓) | Grad. Var. (↓) |
|---|---|---|
| SW-SetDLM $S \leq L$ | $\leq \mathbf{42.02}$ | $\mathbf{2.73 \times 10^{-2}}$ |
| - efficient training | $\leq 54.49$ | $7.50 \times 10^{-2}$ |
| - causal mask | $\leq 46.00$ | $15.01 \times 10^{-2}$ |

MDLMs focusing on masking noise (Lou et al., 2023; Shi et al., 2024; Sahoo et al., 2024), with two key improvements: (1) prior diffusion language models use order-agnostic ordering schedules, whereas we employ position-dependent ordering schedules that tighten the ELBO; (2) our causal diffusion architecture enables KV caching and variable-length generation, which prior approaches do not support. Although Kim et al. (2025b); Seo et al. (2025); Shu et al. (2026) explore non-uniform generation orders, they rely on heuristic ordering rather than our principled per-token noise parameterization, and their bidirectional transformer precludes flexible-length generation and KV caching. Esoteric Language Models (Sahoo et al., 2025b) also interpolate between diffusion and AR, but do so using an order-agnostic diffusion stage before left-to-right AR completion, rather than defining a diffusion process with a left-to-right generation bias. While they also use a causal architecture for KV caching, their training objective applies supervision only at randomly masked positions, resulting in sparser training signals than our objective, and they do not explore variable-length generation. WeDLM (Liu et al., 2025a) adopts a causal transformer architecture for KV caching but does not support variable-length generation or tunable interpolation between diffusion and AR.

*Table 6.* Effect of efficient training and KV caching on GSM8K performance. Decoding throughput (`Tput`) is measured in tokens / sec on an `H100 80GB` GPU. We report mean ± standard deviation for 200 samples.

| | 0-shot pass@1 (↑) | Tput (↑) |
|---|---|---|
| SW-SetDLM $S \leq L$ | **26.61** | $40.83_{\pm 0.38}$ |
| - eff. training, causal mask | 6.37 | $24.48_{\pm 3.82}$ |

**Comparison to Block Diffusion** Block diffusion models (Arriola et al., 2025a) (also known as semi-autoregressive models (Han et al., 2022; 2023)) unmask tokens in strict left-to-right blocks, which sacrifices infilling flexibility and restricts parallelism, as each block must be fully decoded before advancing. In contrast, we support decoding in sliding-window sets. While Arriola et al. (2025a) update KV caches once a block is fully decoded, we update the KV cache after every inference step for greater efficiency.

**Comparison to FlexMDM** FlexMDM (Kim et al., 2025a) also uses any-order autoregression for flexible-length diffusion sampling. However, it introduces separate schedules for masked token insertion and unmasking, requires a separate insertion network to predict inserted tokens at each step, and uses a bidirectional transformer to parameterize token likelihoods, precluding KV caching. In contrast, set diffusion explicitly models generation order, supports flexible-length generation without an insertion network while adhering to a user-specified inference budget, and enables KV caching.

**Comparison to Any-Order Autoregressive Models** Our objective is related to the AO-ARM (Uria et al., 2014), whose objective is equivalent to fully non-AR diffusion (Hoogeboom et al., 2021; Ou et al., 2024; Uria et al., 2014; Shih et al., 2022). Unlike prior AO-ARMs which optimize over uniform orders and sample a single trajectory step during training (Uria et al., 2014; Hoogeboom et al., 2021; Ou et al., 2024), SW-SetDLMs use non-uniform orders and efficiently predict all token likelihoods simultaneously. While Shih et al. (2022) allow non-uniform $\pi(\sigma)$, their method cannot flexibly interpolate ordering distributions and uses a bidirectional transformer, preventing KV caching and variable-length inference. Methods that learn orderings via variational inference (Wang et al., 2025c) or confidence-based unmasking policies (Kim et al., 2026) require complex optimization to discover effective orderings, whereas we impose a structured prior over $\pi(\sigma)$ motivated by the success of block-wise generation (Arriola et al., 2025a).

**Limitations** Although the gap between set diffusion and autoregression has narrowed, there is still a tradeoff between accuracy and parallelization in tuning the position-offset ordering schedules. The ideal schedule can be hand-tuned based on user preferences and/or parallelization capabilities of the inference hardware, or learned throughout training (Kim et al., 2026; Wang et al., 2025c).

**Conclusion** This work introduces set diffusion, a framework for interpolating between AR and diffusion decoding by tuning generation orders rather than fixed block structure. Set diffusion supports flexible-length generation and KV caching with a causal diffusion architecture, achieving state-of-the-art speed-quality tradeoffs across benchmarks.

## Impact Statement

This paper presents work whose goal is to advance the field of Machine Learning. There are many potential societal consequences of our work, none which we feel must be specifically highlighted here.

## Acknowledgements

This work was partially funded by the National Science Foundation under award CAREER 2145577, and by the National Institute of Health under award MIRA R35GM151243. Marianne Arriola is supported by a NSF Graduate Research Fellowship under award DGE-2139899 and a Hopper-Dean/Bowers CIS Deans Excellence Fellowship. Marianne Arriola thanks Yair Schiff and Gilad Turok for feedback on paper figures.

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

# Contents

## A. Latent-Order NELBO for Set Factorization

We derive a latent-order NELBO for autoregressive factorizations over token sets (defined in Def. 3.1):

$$-\log p_\theta(\mathbf{x}) = -\log \sum_\sigma p_\theta(\mathbf{x}, \sigma) \tag{11}$$

$$= -\log \sum_\sigma q(\sigma \mid \mathbf{x}) \frac{p_\theta(\mathbf{x}, \sigma)}{q(\sigma \mid \mathbf{x})}$$

$$= -\log \mathbb{E}_{\sigma \sim q(\cdot \mid \mathbf{x})} \left[ \frac{p_\theta(\mathbf{x}, \sigma)}{q(\sigma \mid \mathbf{x})} \right]$$

$$\leq -\mathbb{E}_{\sigma \sim q(\cdot \mid \mathbf{x})} \left[ \log p_\theta(\mathbf{x}, \sigma) - \log q(\sigma \mid \mathbf{x}) \right]$$

$$= -\mathbb{E}_{\sigma \sim q(\cdot \mid \mathbf{x})} \left[ \log p_\theta(\mathbf{x} \mid \sigma) + \log p_\theta(\sigma) - \log q(\sigma \mid \mathbf{x}) \right]$$

$$= -\mathbb{E}_{\sigma \sim q(\cdot \mid \mathbf{x})} \left[ \sum_{n=1}^N \log p_\theta(\mathbf{x}^{\sigma_n} \mid \mathbf{x}^{\sigma_{<n}}) \right] + D_{\mathrm{KL}} \left[ q(\sigma \mid \mathbf{x}) \parallel p_\theta(\sigma) \right] \tag{12}$$

Under the assumption that orderings are independent of the data $\mathbf{x}$ and model parameters $\theta$, we accordingly choose the prior $p(\sigma) \equiv q(\sigma)$, and the KL term vanishes:

$$-\log p_\theta(\mathbf{x}) \leq -\mathbb{E}_{\sigma \sim q(\sigma)} \left[ \sum_{n=1}^N \log p_\theta(\mathbf{x}^{\sigma_n} \mid \mathbf{x}^{\sigma_{<n}}) \right] \tag{13}$$

The orderings $\sigma$ may be sampled from an order policy $\pi(\sigma)$, such as the position-offset schedules in Section 4.2. Therefore, we choose the variational distribution and the ordering prior to be the same fixed distribution over generation orders where $q(\sigma) \equiv \pi(\sigma)$.

Set diffusion further models each conditional likelihood $p_\theta(\mathbf{x}^{\sigma_n} \mid \mathbf{x}^{<\sigma_n})$ using diffusion. If we denote the diffusion NELBO for the $n$-th token set as $-\log p_\theta(\mathbf{x}^{\sigma_n} \mid \mathbf{x}^{<\sigma_n}) \leq \mathcal{L}(\mathbf{x}^{\sigma_n}, \mathbf{x}^{<\sigma_n}; \theta)$ then combining this conditional NELBO with (13) yields

$$-\log p_\theta(\mathbf{x}) \leq \mathcal{L}_{\mathrm{SD}}(x; \theta) := \mathbb{E}_{\sigma \sim \pi} \sum_{n=1}^N \mathcal{L}(\mathbf{x}^{\sigma_n}, \mathbf{x}^{<\sigma_n}; \theta). \tag{14}$$

At inference time, the sampler may use a different order policy $\pi(\sigma)$, such as a confidence-based or hardware-aware policy, in determining the decoding trajectory.

## B. Relating the Set Diffusion and Masked Diffusion NELBO

The any-order autoregressive objective has been shown to be equivalent to the masked diffusion modeling objective (Hoogeboom et al., 2021; Ou et al., 2024) under uniform orderings. Below, we show that we can relate the masked diffusion and any-order autoregressive objective similarly under arbitrary ordering distributions:

$$-\log p_\theta(\mathbf{x}) \leq -\mathbb{E}_{\sigma \sim q} \left[ \sum_{n=1}^N \log p_\theta(\mathbf{x}^{\sigma_n} \mid \mathbf{x}^{\sigma_{<n}}) \right] \tag{15}$$

$$= -\mathbb{E}_{\sigma \sim q} N \cdot \mathbb{E}_{n \sim \mathcal{U}(\{1,\ldots,N\})} \log p_\theta(\mathbf{x}^{\sigma_n} \mid \mathbf{x}^{\sigma_{<n}})$$

$$= -N \cdot \mathbb{E}_{n \sim \mathcal{U}(\{1,\ldots,N\})} \mathbb{E}_{\sigma_{<n} \sim q} \sum_{k \in \sigma_{\geq n}} q(\sigma_n = k \mid \sigma_{<n}) \log p_\theta(\mathbf{x}^k \mid \mathbf{x}^{\sigma_{<n}}) \tag{16}$$

We recover the masked diffusion objective under uniform orderings $\pi(\sigma) \equiv \mathcal{U}(\{1,\ldots,N\})$:

$$-\log p_\theta(\mathbf{x}) \leq -N \cdot \mathbb{E}_{n \sim \mathcal{U}(\{1,\ldots,N\})} \mathbb{E}_{\sigma \sim q} \frac{1}{N - n + 1} \sum_{k \in \sigma_{\geq n}} \log p_\theta(\mathbf{x}^k \mid \mathbf{x}^{\sigma_{<n}}) \tag{17}$$

$$= -\mathbb{E}_{n \sim \mathcal{U}(\{1,\ldots,N\})} \mathbb{E}_{\sigma \sim q} \frac{1}{1 - \frac{n-1}{N}} \sum_{k \in \sigma_{\geq n}} \log p_\theta(\mathbf{x}^k \mid \mathbf{x}^{\sigma_{<n}}) \tag{18}$$

where $\frac{1}{1-\frac{n-1}{N}}$ is a coefficient that normalizes by the number of predicted tokens, equivalent to $\frac{1}{1-\alpha_t}$ in the MDLM objective, where $\alpha_t$ corresponds to the univariate noise schedule. Normalizing by $\frac{1}{1-\frac{n-1}{N}}$ (the realized prediction ratio) rather than $\frac{1}{1-\alpha_t}$ (the expected prediction ratio) reduces variance, since it exactly matches the ratio of predicted tokens in each sample rather than approximating it in expectation.

When we design $\pi(\sigma)$ to bias toward certain orderings, then $q(\sigma_n = k \mid \sigma_{<n})$ places most of its mass on a small subset of tokens. Thus under high ordering bias, optimizing the masked diffusion objective is inefficient as gradients are computed using the predictions of only a few tokens in a mini-batch (Arriola et al., 2025a).

## C. Relating the Block Diffusion and Set Diffusion NELBO

We show that the set diffusion likelihood (2) can recover the block diffusion likelihood (Arriola et al., 2025a). Assume a block-autoregressive factorization over $B$ token blocks. Let $N_b$ denote the number of sets in block index $b \in \{1, \ldots, B\}$.

$$-\log p_\theta(\mathbf{x}) \leq -\mathbb{E}_{\sigma \sim q(\cdot \mid \mathbf{x})} \left[ \sum_{n=1}^{N} \log p_\theta(\mathbf{x}^{\sigma_n} \mid \mathbf{x}^{\sigma_{<n}}) \right] \tag{19}$$

$$= -\sum_{b=1}^{B} \mathbb{E}_{\sigma \sim q(\cdot \mid \mathbf{x})} \left[ \sum_{n=1}^{N_b} \log p_\theta(\mathbf{x}^{\sigma_n} \mid \mathbf{x}^{\sigma_{<n}}, \mathbf{x}^{<b}) \right] \tag{20}$$

$$= -\sum_{b=1}^{B} N_b \cdot \mathbb{E}_{\sigma \sim q(\cdot \mid \mathbf{x})} \mathbb{E}_{n \sim \mathcal{U}(\{1, \ldots, N_b\})} \log p_\theta(\mathbf{x}^{\sigma_n} \mid \mathbf{x}^{\sigma_{<n}}, \mathbf{x}^{<b}) \tag{21}$$

$$= -\sum_{b=1}^{B} N_b \cdot \mathbb{E}_{n \sim \mathcal{U}(\{1, \ldots, N_b\})} \mathbb{E}_{\sigma_{<n} \sim q(\cdot \mid \mathbf{x})} \sum_{k \in \sigma_{\geq n}} q(\sigma_n = k \mid \sigma_{<n}) \log p_\theta(\mathbf{x}^k \mid \mathbf{x}^{\sigma_{<n}}, \mathbf{x}^{<b}) \tag{22}$$

$$= -\sum_{b=1}^{B} \mathbb{E}_{n \sim \mathcal{U}(\{1, \ldots, N_b\})} \mathbb{E}_{\sigma_{<n} \sim q(\cdot \mid \mathbf{x})} \frac{1}{1 - \frac{n-1}{N_b}} \sum_{k \in \sigma_{\geq n}} \log p_\theta(\mathbf{x}^k \mid \mathbf{x}^{\sigma_{<n}}, \mathbf{x}^{<b}) \tag{23}$$

where $\frac{1}{1-\frac{n-1}{N_b}}$ is a coefficient that normalizes by the number of predicted tokens in a block, equivalent to $\frac{1}{1-\alpha_t^b}$ in the block diffusion objective, where $\alpha_t^b$ is a univariate noise schedule applied to tokens in the $b$-th block. Normalizing by $\frac{1}{1-\frac{n-1}{N_b}}$ (the realized prediction ratio) rather than $\frac{1}{1-\alpha_t^b}$ (the expected prediction ratio) reduces variance, since it exactly matches the ratio of predicted tokens in each sample rather than approximating it in expectation.

## D. Relating the Set Diffusion and AO-ARM NELBO

We show that the singleton-set SW-SetDLM objective recovers the any-order autoregressive modeling (AO-ARM) objective (Uria et al., 2014) when the ordering distribution is uniform over permutations. More generally, it yields a non-uniform AO-ARM objective when $\pi(\sigma)$ is allowed to place structured mass on particular generation orders.

For singleton token sets where $N = L$, the set diffusion NELBO in (3) becomes

$$-\log p_\theta(\mathbf{x}) \leq \mathcal{L}_{\text{SD}}(\mathbf{x}; \theta) := -\mathbb{E}_{\sigma \sim \pi} \sum_{n=1}^{L} \log p_\theta(\mathbf{x}^{\sigma_n} \mid \mathbf{x}^{\sigma_{<n}}). \tag{24}$$

Thus, training samples a full ordering over token positions and predicts each token conditioned on the tokens that precede it in that ordering. Prior AO-ARM and masked-diffusion objectives use a masked-diffusion-style estimator that samples an ordering step and averages over the possible next positions (Uria et al., 2014; Hoogeboom et al., 2021; Shih et al., 2022):

$$-\log p_\theta(\mathbf{x}) \leq -N \, \mathbb{E}_{n, \sigma_{<n}} \sum_{k \in \sigma_{\geq n}} \pi(\sigma_n = k \mid \sigma_{<n}) \log p_\theta(\mathbf{x}^k \mid \mathbf{x}^{\sigma_{<n}}) \tag{25}$$

Under biased $\pi$, this estimator is inefficient: $\pi(\sigma_n = k \mid \sigma_{<n})$ concentrates on only a few remaining positions, so each update receives gradient signal from a small subset of token predictions.

# E. Multivariate Masked Diffusion NELBO

We derive the masked diffusion NELBO under a multivariate noise schedule $\boldsymbol{\alpha}_t \in [0,1]^L$ for $L$ tokens. Each $\alpha_t^\ell = \alpha^\ell(t)$ is the noise schedule for the $\ell$-th token monotonically decreasing in $t$ corresponding to the probability that the $\ell$-th token remains unmasked at time $t$ (Sahoo et al., 2024; Lou et al., 2023).

$$-\log p_\theta(\mathbf{x}^{1:L}) = -\log \mathbb{E}_{q(\mathbf{z}_{1:T}^{1:L}|\mathbf{x}^{1:L})} \frac{p_\theta(\mathbf{z}_{1:T}^{1:L})}{q(\mathbf{z}_{1:T}^{1:L}|\mathbf{x}^{1:L})} \tag{26}$$

$$= -\log \mathbb{E}_q \frac{p_\theta(\mathbf{z}_T^{1:L}) \prod_{t=1}^{T} p_\theta(\mathbf{z}_{t-1}^{1:L}|\mathbf{z}_t^{1:L})}{\prod_{t=1}^{T} q(\mathbf{z}_t^{1:L}|\mathbf{z}_{t-1}^{1:L})} \tag{27}$$

$$\leq \mathbb{E}_q \Big[ D_{\mathrm{KL}}(q(\mathbf{z}_T^{1:L}|\mathbf{x}^{1:L}) \,\|\, p_\theta(\mathbf{z}_T^{1:L}))$$

$$+ \sum_{t=2}^{T} D_{\mathrm{KL}}(q(\mathbf{z}_{t-1}^{1:L}|\mathbf{z}_t^{1:L},\mathbf{x}^{1:L}) \,\|\, p_\theta(\mathbf{z}_{t-1}^{1:L}|\mathbf{z}_t^{1:L}))$$

$$- \log p_\theta(\mathbf{x}^{1:L}|\mathbf{z}_1^{1:L}) \Big] \tag{28}$$

We adopt the MDLM NELBO from Sahoo et al. (2024) that simplifies the reverse posterior $q(\mathbf{z}_s^\ell|\mathbf{z}_t^\ell,\mathbf{x}^\ell)$ for absorbing state diffusion. Below, we summarize the derivation from Sahoo et al. (2024). We now simplify the diffusion loss term by marginalizing over all values that can be taken by $(\mathbf{z}_s^\ell, \mathbf{z}_t^\ell)$ for token $\ell \in \{1,\dots,L\}$, which are $(\mathbf{m},\mathbf{m}), (\mathbf{x}^\ell,\mathbf{m}), (\mathbf{x}^\ell,\mathbf{x}^\ell)$. We note that the state $(\mathbf{m},\mathbf{x}^\ell)$ is impossible, because tokens are never unmasked after having been masked in the forward process $q$.

**Case $\mathbf{z}_s^\ell = \mathbf{m}, \mathbf{z}_t^\ell = \mathbf{m}$:**   Observe that

$$\log \frac{q(\mathbf{z}_s^\ell = \mathbf{m}|\mathbf{z}_t^\ell = \mathbf{m},\mathbf{x}^\ell)}{p_\theta(\mathbf{z}_s^\ell = \mathbf{m}|\mathbf{z}_t^\ell = \mathbf{m})} = \log \frac{(1-\alpha_s)(\alpha_t p_\theta(\mathbf{m}|\mathbf{z}_t^\ell) + (1-\alpha_t))}{(1-\alpha_t)(\alpha_s p_\theta(\mathbf{m}|\mathbf{z}_t^\ell) + (1-\alpha_s))} = 0 \tag{29}$$

where the last equality holds because $p_\theta(\mathbf{m}|\mathbf{z}_t^\ell) = 0$.

**Case $\mathbf{z}_s^\ell = \mathbf{x}^\ell, \mathbf{z}_t^\ell = \mathbf{m}$:**   Observe that

$$\log \frac{q(\mathbf{z}_s^\ell = \mathbf{x}^\ell|\mathbf{z}_t^\ell = \mathbf{m},\mathbf{x}^\ell)}{p_\theta(\mathbf{z}_s^\ell = \mathbf{x}^\ell|\mathbf{z}_t^\ell = \mathbf{m})} = \log \frac{\alpha_t p_\theta(\mathbf{m}|\mathbf{z}_t^\ell) + (1-\alpha_t)}{(1-\alpha_t)p_\theta(\mathbf{x}^\ell|\mathbf{z}_t^\ell)} = -\log p_\theta(\mathbf{x}^\ell|\mathbf{z}_t^\ell) \tag{30}$$

where the last equality holds because $p_\theta(\mathbf{m}|\mathbf{z}_t^\ell) = 0$.

**Case $\mathbf{z}_s^\ell = \mathbf{x}^\ell, \mathbf{z}_t^\ell = \mathbf{x}^\ell$:**   The ratio becomes 0 because $p_\theta(\mathbf{z}_s^\ell = \mathbf{x}^\ell|\mathbf{z}_t^\ell = \mathbf{x}^\ell) = q(\mathbf{z}_s^\ell = \mathbf{x}^\ell|\mathbf{z}_t^\ell = \mathbf{x}^\ell,\mathbf{x}^\ell)$.

$$= \prod_{\ell=1}^{L} \big[ q(\mathbf{z}_s^\ell = \mathbf{x}|\mathbf{z}_t^\ell = \mathbf{m},\mathbf{x}^\ell) \big] \log \prod_{\ell=1}^{L} \frac{q(\mathbf{z}_s^\ell = \mathbf{x}|\mathbf{z}_t^\ell = \mathbf{m},\mathbf{x}^\ell)}{p_\theta(\mathbf{z}_s^\ell = \mathbf{x}|\mathbf{z}_t^{1:L} = \mathbf{m})} \tag{31}$$

$$+ \prod_{\ell=1}^{L} \big[ q(\mathbf{z}_s^\ell = \mathbf{m}|\mathbf{z}_t^\ell = \mathbf{m},\mathbf{x}^\ell) \big] \log \prod_{\ell=1}^{L} \frac{q(\mathbf{z}_s^\ell = \mathbf{m}|\mathbf{z}_t^\ell = \mathbf{m},\mathbf{x}^\ell)}{p_\theta(\mathbf{z}_s^\ell = \mathbf{x}|\mathbf{z}_t^{1:L} = \mathbf{m})} \tag{32}$$

$$= T \Bigg[ \sum_{\ell=1}^{L} \frac{\alpha_s^\ell - \alpha_t^\ell}{1-\alpha_t^\ell} \log \frac{\alpha_t^\ell p_\theta(\mathbf{x}^\ell = \mathbf{m} \mid \mathbf{z}_t^{1:L}) + (1-\alpha_t^\ell)}{(1-\alpha_t^\ell)p_\theta(\mathbf{x}^\ell \neq \mathbf{m} \mid \mathbf{z}_t^{1:L})}$$

$$+ \frac{1-\alpha_s^\ell}{1-\alpha_t^\ell} \log \frac{(1-\alpha_s^\ell)(\alpha_t^\ell p_\theta(\mathbf{x}^\ell = \mathbf{m} \mid \mathbf{z}_t^{1:L}) + (1-\alpha_t^\ell))}{(1-\alpha_t^\ell)(\alpha_s^\ell p_\theta(\mathbf{x}^\ell = \mathbf{m} \mid \mathbf{z}_t^{1:L}) + (1-\alpha_s^\ell))} \Bigg] \tag{33}$$

$$= \mathbb{E}_{t\sim(0,1]} \mathbb{E}_q T \Bigg[ \sum_{\ell=1}^{L} \frac{\alpha_s^\ell - \alpha_t^\ell}{1-\alpha_t^\ell} \log p_\theta(\mathbf{x}^\ell \mid \mathbf{z}_t^{1:L}) \Bigg] \tag{34}$$

$$= \mathbb{E}_{t\sim(0,1]} \mathbb{E}_q \Bigg[ \sum_{\ell=1}^{L} \frac{\dot{\alpha}_t^{\,\ell}}{1-\alpha_t^\ell} \log p_\theta(\mathbf{x}^\ell \mid \mathbf{z}_t^{1:L}) \Bigg] \tag{35}$$

### E.1. Diffusion NELBO Depends on Multivariate Noise Schedules

We will show that the diffusion NELBO below in Eq. 36 is dependent on the noise schedule $\boldsymbol{\alpha}_t^{1:L}$ when it is multi-dimensional. This finding enables us to craft the noise schedule $\boldsymbol{\alpha}_t^{1:L}$ that is a tighter approximation of the true likelihood.

To do so, we will write the NELBO as a line integral $\int_a^b \mathbf{f}(\mathbf{r}(t)) \cdot \frac{d}{dt} \mathbf{r}(t) dt$ for a vector field $\mathbf{f}$ and a trajectory $\mathbf{r}$. We will show that the trajectory $\mathbf{r}(t)$ is parameterized by the noise schedule $\boldsymbol{\nu}(t)$, and thus the NELBO depends on the noise schedule if it is multi-dimensional as in Gaussian diffusion (Sahoo et al., 2023).

$$\mathcal{L}_{\text{NELBO}}^{\infty} = \mathbb{E}_q \int_{t=0}^{t=1} \frac{\dot{\boldsymbol{\alpha}}_t^{1:L}}{1 - \boldsymbol{\alpha}_t^{1:L}} \odot \log p_\theta(\mathbf{x}^{1:L} \mid \mathbf{z}_t^{1:L}) \, dt \tag{36}$$

$$= \mathbb{E}_q \int_{t=0}^{t=1} \log p_\theta(\mathbf{x}^{1:L} \mid \mathbf{z}_t^{1:L}) \odot \nabla_t [\log(1 - \boldsymbol{\alpha}_t^{1:L})] \, dt \tag{37}$$

We substitute $\boldsymbol{\mu}_\theta(\mathbf{z}_t) = \log p_\theta(\mathbf{x}^{1:L} \mid \mathbf{z}_t^{1:L})$, $\boldsymbol{\nu}(t) = \log(1 - \boldsymbol{\alpha}_t^{1:L})$:

$$\mathcal{L}_{\text{NELBO}}^{\infty} = \mathbb{E}_q \int_{t=0}^{t=1} \boldsymbol{\mu}_\theta(\mathbf{z}_t) \odot \nabla_t \boldsymbol{\nu}(t) \, dt \tag{38}$$

Since $\boldsymbol{\nu}(t)$ is monotonically increasing in $t$, we can express $t$ as an inverse function of $\boldsymbol{\nu}$: $t = \boldsymbol{\nu}^{-1}(\boldsymbol{\nu}(t))$ for some $\boldsymbol{\nu}^{-1}$. We perform a change of variables and set $\bar{\boldsymbol{\mu}}_\theta(\mathbf{x}_{\boldsymbol{\nu}(t)}) \equiv \boldsymbol{\mu}_\theta(\mathbf{x}_{\boldsymbol{\nu}^{-1}(\boldsymbol{\nu}(t))})$. For all $t$, we can write $\boldsymbol{\mu}_t$ as $\boldsymbol{\mu}_{\boldsymbol{\nu}(t)}$ and have $\bar{\boldsymbol{\mu}}_\theta(\mathbf{x}_{\boldsymbol{\nu}(t)}) \equiv \boldsymbol{\mu}_\theta(\mathbf{z}_t)$. Thus, we can rewrite the NELBO as:

$$\mathcal{L}_{\text{NELBO}}^{\infty} = \mathbb{E}_q \int_{t=0}^{t=1} \bar{\boldsymbol{\mu}}_\theta(\mathbf{x}_{\boldsymbol{\nu}(t)}) \odot \frac{d}{dt} \boldsymbol{\nu}(t) \, dt \tag{39}$$

The above NELBO is a line integral with $\mathbf{f}(\mathbf{r}(t)) \equiv \bar{\boldsymbol{\mu}}_\theta(\mathbf{x}_{\boldsymbol{\nu}(t)})$ and $\mathbf{r}(t) \equiv \boldsymbol{\nu}(t)$. Thus, the NELBO is dependent on the noise schedule $\boldsymbol{\nu}(t) = \log(1 - \boldsymbol{\alpha}_t^{1:L})$ and we can design $\boldsymbol{\alpha}_t^{1:L}$ so that the NELBO is a tighter approximation of the true likelihood.

## F. Sliding-Window Set Diffusion NELBO Tightness

We will establish Prop. 4.1 by showing that as the decoding width $w \to \frac{1}{L}$, the NELBO tends to an autoregressive objective. In the limit, this holds true with equality and the NELBO is tight.

Let $\pi_w$ denote the ordering distribution induced by the position-offset schedule. Recall that the offsets are evenly spaced and increasing in token index:

$$a^\ell = \frac{\ell - 1}{L - 1}(1 - w), \qquad \ell \in \{1, \ldots, L\}. \tag{40}$$

The reveal time $R^\ell$ for token $\ell$ is supported on the active generation interval $[a^\ell, a^\ell + w]$. When $w = \frac{1}{L}$, the offsets become $a^\ell = \frac{\ell - 1}{L}$, and therefore the active generation interval becomes $\left[\frac{\ell - 1}{L}, \frac{\ell}{L}\right]$ for token $\ell$. Hence, reveal times respect token order $R^\ell \leq R^j$ for $\ell < j$. Further, no two tokens are selected simultaneously since their generation intervals are disjoint. As a result, the sets are singletons, $N = L$, and the only sampled order is the identity left-to-right order.

Starting from the set-diffusion negative log-likelihood in (11), we obtain

$$\lim_{w \to 1/L} \left\{ -\log \mathbb{E}_{\sigma \sim \pi_w} \left[ \prod_{n=1}^{N} p_\theta(\mathbf{x}^{\sigma_n} \mid \mathbf{x}^{<\sigma_n}) \right] \right\} = -\log \prod_{n=1}^{L} p_\theta(\mathbf{x}^n \mid \mathbf{x}^{<n}) \tag{41}$$

$$= -\sum_{n=1}^{L} \log p_\theta(\mathbf{x}^n \mid \mathbf{x}^{<n}). \tag{42}$$

At $w = \frac{1}{L}$, the convergence above holds with equality because $\pi_{1/L}$ is a point mass on the identity singleton order. Thus the SW-SetDLM objective recovers the autoregressive NLL, and the NELBO is tight.

| | Interpolating quality | Any-length gen. | Infilling flexibility | Full KV-caching | Token-efficient training |
|---|:---:|:---:|:---:|:---:|:---:|
| Autoregression | | ✓ | | ✓ | |
| Bidirectional masked diffusion (Sahoo et al., 2024; Lou et al., 2023; Shi et al., 2024) | | | ✓ | | |
| Block diffusion (Arriola et al., 2025a) | ✓ | ✓ | | | |
| Any-order autoregression (Pannatier et al., 2024; Hoogeboom et al., 2021) | | ✓ | ✓ | ✓ | ✓ |
| **Set diffusion (Ours)** | ✓ | ✓ | ✓ | ✓ | ✓ |

*Table 7.* Comparison between set diffusion and related language modeling paradigms.

## G. Comparison with Related Language Modeling Paradigms

In Table 7, we provide a concise comparison between set diffusion models and other language modeling paradigms.

## H. Tuning Position-Offset Schedules

We compare position-offset ordering schedules with alternative modeling paradigms, such as block diffusion (Arriola et al., 2025a), by controlling for the expected inference prediction budget (Def. 4.2). This quantity corresponds to the expected number of tokens eligible for prediction at a given ordering time, averaged over the generation trajectory.

A token is "eligible" at ordering time $\tau$ if it has not yet been selected for generation and $\tau$ lies in its active generation interval. For a reveal-time variable $R^\ell$, the probability that the $\ell$-th token has not yet been selected is the survival probability $\Pr(R^\ell > \tau) = 1 - \alpha_\tau^\ell$. Thus, the expected inference prediction budget is:

$$\bar{C} = \int_0^1 \sum_{\ell=1}^{L} \Pr(R^\ell > \tau) \cdot \mathbf{1}\{\dot{\alpha}_\tau^\ell \neq 0\} \, \mathrm{d}\tau \tag{43}$$

$$= \int_0^1 \sum_{\ell=1}^{L} (1 - \alpha_\tau^\ell) \cdot \mathbf{1}\{\dot{\alpha}_\tau^\ell \neq 0\} \, \mathrm{d}\tau \tag{44}$$

The offset-linear schedule in Eq. 7 tunes generation only through the interval width $w$. This couples the expected prediction budget to the amount of interval overlap. In particular, for the linear schedule,

$$\bar{C} = \frac{Lw}{2}, \qquad \text{so} \qquad w = \frac{2\bar{C}}{L}. \tag{45}$$

Thus, for small target budgets $\bar{C}$, an offset-linear schedule requires narrow active generation intervals, which may limit decoding flexibility. Non-overlapping intervals impose deterministic ordering constraints: if $a^\ell + w \leq a^j$ for $\ell < j$, then token $\ell$ is always revealed before token $j$. Therefore, tuning only $w$ under the offset-linear schedule reduces the amount of overlap between eligible positions, limiting the ordering flexibility available at that fixed budget.

In practice, we use a shaped variant of the position-offset schedule. Within each active generation interval, we define

$$\alpha_\tau^\ell = \begin{cases} 0, & \tau \leq a^\ell, \\ \left(\frac{\tau - a^\ell}{w}\right)^k, & a^\ell < \tau < a^\ell + w, \\ 1, & \tau \geq a^\ell + w, \end{cases} \tag{46}$$

where $w$ controls the amount of interval overlap and $k > 0$ controls the shape of the reveal-time distribution within each interval. Smaller values of $k$ concentrate reveal times closer to the beginning of the active generation interval.

For this shaped schedule, Eq. 8 has a closed form. For each position $\ell$, we make the change of variables

$$u = \frac{\tau - a^\ell}{w}, \qquad d\tau = w\,du. \tag{47}$$

When $\tau = a^\ell$, we have $u = 0$, and when $\tau = a^\ell + w$, we have $u = 1$. Therefore,

$$\bar{C} = \sum_{\ell=1}^{L} \int_{a^\ell}^{a^\ell+w} \left[1 - \left(\frac{\tau - a^\ell}{w}\right)^k\right] d\tau \tag{48}$$

$$= w \sum_{\ell=1}^{L} \int_{0}^{1} (1 - u^k)\,du \tag{49}$$

$$= w \sum_{\ell=1}^{L} \left[u - \frac{u^{k+1}}{k+1}\right]_{0}^{1} \tag{50}$$

$$= w \sum_{\ell=1}^{L} \left(1 - \frac{1}{k+1}\right) \tag{51}$$

$$= \frac{Lwk}{k+1}. \tag{52}$$

Therefore, for a target prediction budget $\bar{C}$ and a chosen interval width $w$, we set

$$k = \frac{\bar{C}}{Lw - \bar{C}}, \tag{53}$$

provided that $Lw > \bar{C}$. To obtain wider active intervals than the offset-linear schedule, we choose $w > 2\bar{C}/L$, which implies $k < 1$. This increases overlap between active generation intervals while preserving the same expected prediction budget by setting $k = \bar{C}/(Lw - \bar{C})$. In this regime, reveal times are biased toward the beginning of each active generation interval, making tokens more likely to be revealed soon after becoming eligible while retaining greater ordering flexibility. Figure 5 illustrates how shaped schedules preserve the same expected prediction budget as offset-linear schedules while enabling more flexible any-order decoding.

We choose $w$ by specifying a desired lookahead. The lookahead of token $\ell$ is the number of later positions $j > \ell$ that can become eligible for prediction before token $\ell$ is necessarily generated. A later token $j$ can become eligible before token $\ell$'s active generation interval ends whenever $a^j < a^\ell + w$. Since the offsets are evenly spaced,

$$a^\ell = \frac{\ell - 1}{L - 1}(1 - w). \tag{54}$$

The spacing between adjacent offsets is

$$\Delta a = a^{\ell+1} - a^\ell \tag{55}$$

$$= \frac{\ell}{L-1}(1-w) - \frac{\ell-1}{L-1}(1-w) \tag{56}$$

$$= \frac{1-w}{L-1}. \tag{57}$$

Writing $j = \ell + d$, we have $a^j = a^\ell + d\Delta a$. Therefore,

$$a^{\ell+d} < a^\ell + w \tag{58}$$

$$a^\ell + d\Delta a < a^\ell + w \tag{59}$$

$$d < \frac{w(L-1)}{1-w}. \tag{60}$$

For large $L$, the approximate maximum lookahead induced by interval width $w$ is

$$N_{\text{ov}} \approx \frac{wL}{1-w}. \tag{61}$$

Equivalently, for a desired lookahead $N_{\text{ov}}$, we choose

$$w \approx \frac{N_{\text{ov}}}{L + N_{\text{ov}}}, \qquad k = \frac{\bar{C}}{Lw - \bar{C}}. \tag{62}$$

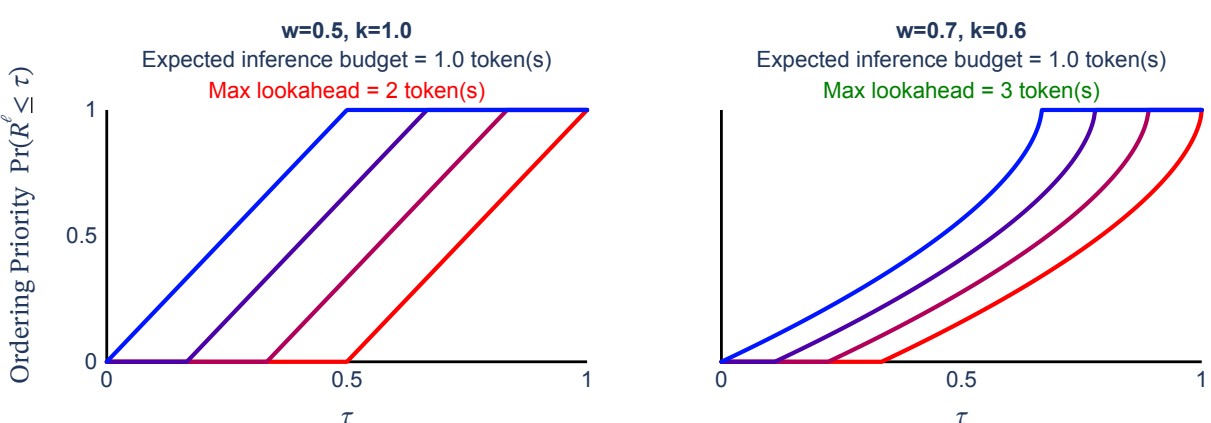

*Figure 5.* Effect of tuning the position-offset ordering schedule parameters $w, k$ under a fixed expected inference prediction budget $\bar{C} = 1$, matched to a BD3LM block size of 2. The schedules induce different maximum lookahead values, i.e., the maximum number of later tokens that can become eligible for prediction ahead of a given token.

## I. Expected Inference Prediction Budget

**Order-agnostic diffusion.** Assume a linear noise schedule $\alpha_\tau = \tau$. For order-agnostic diffusion (e.g., MDLM; Sahoo et al. (2024); Shi et al. (2024); Ou et al. (2024)), the expected number of tokens that can be predicted is:

$$\bar{C} = \sum_{\ell=1}^{L} \int_0^1 (1 - \tau) \mathrm{d}\tau = \frac{L}{2} \tag{63}$$

**Block diffusion.** For block diffusion, let $w = S/L$ denote the normalized generation interval width for block size $S$. For token position $\ell \in \{1, \dots, L\}$, let

$$a^\ell = \left\lfloor \frac{\ell - 1}{S} \right\rfloor \frac{S}{L} \tag{64}$$

denote the beginning of the corresponding block interval. Assuming a linear reveal schedule within blocks,

$$\alpha_\tau^\ell = \frac{\tau - a^\ell}{w}, \qquad \tau \in [a^\ell, a^\ell + w], \tag{65}$$

the probability that token $\ell$ has not yet been selected is

$$1 - \alpha_\tau^\ell = 1 - \frac{\tau - a^\ell}{w}. \tag{66}$$

Thus, the expected number of tokens that can be predicted is

$$\bar{C}_{\text{BD}}(S) = \sum_{\ell=1}^{L} \int_{a^\ell}^{a^\ell + w} \left(1 - \frac{\tau - a^\ell}{w}\right) \mathrm{d}\tau. \tag{67}$$

Under the change of variables $u = \tau - a^\ell$,

$$\bar{C}_{\text{BD}}(S) = \sum_{\ell=1}^{L} \int_0^w \left(1 - \frac{u}{w}\right) \, \mathrm{d}u \tag{68}$$

$$= \sum_{\ell=1}^{L} \frac{w}{2} = \sum_{\ell=1}^{L} \frac{S}{2L} = \frac{S}{2}. \tag{69}$$

**Position-offset schedule.** For the position-offset ordering schedule, the reveal CDF within the active generation interval $[a^\ell, a^\ell + w]$ is

$$\alpha_\tau^\ell = \left(\frac{\tau - a^\ell}{w}\right)^k. \tag{70}$$

The expected inference prediction budget is therefore

$$\bar{C}_{\text{offset}}(w, k) = \sum_{\ell=1}^{L} \int_{a^\ell}^{a^\ell + w} \left[1 - \left(\frac{\tau - a^\ell}{w}\right)^k\right] \, \mathrm{d}\tau. \tag{71}$$

Using the closed-form expression derived in (52), we obtain

$$\bar{C}_{\text{offset}}(w, k) = \frac{Lwk}{k+1}. \tag{72}$$

## J. Sampling Generation Trajectories from Position-Offset Schedules

We sample the ordering trajectories used during training by sampling a generation time for each token, inspired by the first-hitting sampler (Zheng et al., 2024). The sampled generation times induce a permutation over token positions, which determines the order in which singleton token sets are predicted during training.

For each token position $\ell \in \{1, \ldots, L\}$, let $R^\ell \in [a^\ell, a^\ell + w]$ denote the first ordering time at which token $\ell$ is selected. We use the position-offset ordering schedule, where $\alpha_\tau^\ell$ denotes the probability that the $\ell$-th token has been revealed by ordering time $\tau$. Hence, the generation-time CDF for $\tau \in [a^\ell, a^\ell + w]$ is:

$$\Pr(R^\ell \leq \tau) = \alpha_\tau^\ell \tag{73}$$

$$= \left(\frac{\tau - a^\ell}{w}\right)^k \tag{74}$$

Given $u^\ell \sim \text{Uniform}(0, 1)$, inverse-CDF sampling gives

$$u^\ell = \left(\frac{R^\ell - a^\ell}{w}\right)^k \tag{75}$$

$$R^\ell = a^\ell + w(u^\ell)^{1/k} \tag{76}$$

After independently sampling $R^1, \ldots, R^L$, we obtain the generation order by sorting generation times in ascending order $\sigma = \text{argsort}(R^1, \ldots, R^L)$. Thus, tokens with smaller ordering times appear earlier in the generation order. Reveal times can also be rounded to a grid with $m$ intervals on $[0, 1]$ as $\tilde{R}^\ell = \frac{1}{m} \arg\min_{i \in \{0, \ldots, m\}} \left|\frac{i}{m} - R^\ell\right|$ to induce a position sequence of non-singleton sets, where each set groups together identical rounded reveal times.

## K. Specialized Attention Masks

We aim to model conditional probabilities $p_\theta(\mathbf{x}^{\sigma_n} \mid \mathbf{z}_t^{\sigma_n}, \mathbf{x}^{\sigma_{<n}})$ for all position sets $\sigma_1, \ldots, \sigma_N$ simultaneously by designing an efficient training algorithm with our transformer backbone. However, modeling all $N$ conditional terms requires processing both the noised sequence $\mathbf{z}_t^{\sigma_n}$ and the conditional context $\mathbf{x}^{\sigma_{<n}}$ for all token sets.

Rather than calling the denoising network $N$ times, we process both sequences simultaneously by concatenating them $\mathbf{x}_{\text{in}} = \mathbf{x}^{\sigma_1} \oplus \cdots \oplus \mathbf{x}^{\sigma_N} \oplus \mathbf{z}_{t_1}^{\sigma_1} \oplus \cdots \oplus \mathbf{z}_{t_N}^{\sigma_N}$ as input to a transformer. We update this sequence $\mathbf{x}_{\text{in}}$ of length $2L$ tokens using a custom attention mask $\mathcal{M} \in \{0,1\}^{2L \times 2L}$ for efficient training. This generalizes the custom attention mask used in block diffusion (Arriola et al., 2025a) to arbitrary-position, arbitrary-length token sets.

The full attention mask is composed of four $L \times L$ smaller attention masks:

$$\mathcal{M} = \begin{bmatrix} \mathcal{M}_{SC} & \mathbf{0} \\ \mathcal{M}_{OSC} & \mathcal{M}_{SD} \end{bmatrix}$$

where $\mathcal{M}_{SD}$ and $\mathcal{M}_{OSC}$ are used to update the representation of $\mathbf{x}_t$ and $\mathcal{M}_{SC}$ is used to update the representation of $\mathbf{x}$. We define these masks as follows:

- $\mathcal{M}_{SD}$ (Set-diagonal mask): Self-attention mask within noised token sets $\mathbf{z}_t^{\sigma_n}$

$$[\mathcal{M}_{SD}]_{ij} = \begin{cases} 1 & \text{if } i, j \text{ are in the same token set} \\ 0 & \text{otherwise} \end{cases}$$

- $\mathcal{M}_{OSC}$ (Offset set-causal mask): Cross-attention to conditional context $\mathbf{x}^{\sigma_{<n}}$

$$[\mathcal{M}_{OSC}]_{ij} = \begin{cases} 1 & \text{if } j \text{ belongs to a token set preceding } i \\ 0 & \text{otherwise} \end{cases}$$

- $\mathcal{M}_{SC}$ (Set-causal mask): Attention mask for updating $\mathbf{x}^{\sigma_n}$

$$[\mathcal{M}_{SC}]_{ij} = \begin{cases} 1 & \text{if } j \text{ belongs to the same token set as } i, \text{ or a token set before } i \\ 0 & \text{otherwise} \end{cases}$$

### K.1. Gradient Variance Estimator

Below, we derive an estimator for gradient variance in SW-SetDLM training. We denote a batch of sequences as $\mathbf{X} = \left[\mathbf{x}^{(1)}, \mathbf{x}^{(2)}, \ldots, \mathbf{x}^{(K)}\right]$ for batch size $K$, with each $\mathbf{x}^{(k)} \overset{\text{iid}}{\sim} q(\mathbf{x})$ and $\mathbf{x}^{(k,\ell)}$ denoting the $\ell$-th token for $\ell \in \{1, \ldots, L\}$. In set diffusion, we factorize the generation over $N$ token sets.

$$\text{Var}_{\mathbf{X},\sigma}\left[\nabla_\theta \mathcal{L}_{\text{SW-SetDLM}}(\mathbf{X}; \theta)\right] = \text{Var}_{\mathbf{X},\sigma}\left[\frac{1}{KL}\sum_{k=1}^{K}\sum_{n=1}^{N} \nabla_\theta \log p_\theta\left(\mathbf{x}^{(k,\sigma_n)} \mid \mathbf{x}^{(k,\sigma_{<n})}\right)\right] \tag{77}$$

$$= \frac{1}{KL}\sum_{k=1}^{K}\sum_{n=1}^{N} \left\|\nabla_\theta \log p_\theta(\mathbf{x}^{(k,\sigma_n)} | \mathbf{x}^{(k,\sigma_{<n})}) - \nabla_\theta \mathcal{L}(\mathbf{X}; \theta)\right\|_2^2 \tag{78}$$

We show that the gradient variance from set diffusion training is lower than that of standard masked diffusion training, which computes gradients using randomly masked tokens (Sahoo et al., 2024; Shi et al., 2024; Ou et al., 2024; Sahoo et al., 2025b; Kim et al., 2025a). In contrast, SW-SetDLMs use all tokens in a batch. The loss for clean tokens $\mathcal{C}^{(k)} = \left\{\mathbf{z}_{t(k)}^{(k,\ell)} \mid \ell \in \{1, \ldots, L\}, \mathbf{z}_{t(k)}^{(k,\ell)} \neq \mathbf{m}\right\}$ is set to 0 following the derivation from (Sahoo et al., 2024; Shi et al., 2024; Ou et al., 2024) (which can be thought of as dropout). Similar to standard dropout, we rescale MDLM NELBO by the inverse percentage of masked tokens $\frac{L}{L-|\mathcal{C}^{(k)}|}$ so that the gradient estimate remains unbiased. The MDLM gradient variance estimator is expressed as:

$$\text{Var}_{\mathbf{X},t}[\nabla_\theta \mathcal{L}_{\text{MDLM}}(\mathbf{X};\theta)]$$

$$= \frac{L}{K} \sum_{k=1}^{K} \sum_{\ell=1}^{L} \left\| \frac{\nabla_\theta \log p_\theta(\mathbf{x}^{(k,\ell)} \mid \mathbf{z}_{t(k)}^{(k,\cdot)})}{L - |\mathcal{C}^{(k)}|} - \nabla_\theta \mathcal{L}(\mathbf{X};\theta) \right\|_2^2$$

$$= \frac{L}{K} \sum_{k=1}^{K} \left[ |\mathcal{C}^{(k)}| \left\| \nabla_\theta \mathcal{L}(\mathbf{X};\theta) \right\|_2^2 \right.$$

$$\left. + \sum_{\ell \notin \mathcal{C}^{(k)}} \left\| \frac{\nabla_\theta \log p_\theta(\mathbf{x}^{(k,\ell)} \mid \mathbf{z}_{t(k)}^{(k,\cdot)})}{L - |\mathcal{C}^{(k)}|} - \nabla_\theta \mathcal{L}(\mathbf{X};\theta) \right\|_2^2 \right]. \tag{79}$$

Even though the gradient contributions from masked tokens are appropriately rescaled, the gradient variance induced by clean tokens (i.e. dropout tokens) is non-zero. Thus, the estimator of MDLM gradient variance is lower-bounded by $\frac{1}{KL} \sum_{k=1}^{K} |\mathcal{C}^{(k)}| \left\| \nabla_\theta \mathcal{L}(\mathbf{X};\theta) \right\|_2^2$.

## L. Experimental Details

For GSM8K and CNN/DailyMail, we use AdamW with weight decay $1e^{-5}$ and $(\beta_1, \beta_2) = (0.9, 0.98)$. For OWT and LM1B, we use AdamW with $(\beta_1, \beta_2) = (0.9, 0.999)$ and no weight decay, following Sahoo et al. (2024); Arriola et al. (2025a).

We apply gradient clipping with a maximum norm of 1.0. We maintain an exponential moving average (EMA) of model parameters with decay rate 0.9999, and use the EMA model for evaluation. For decoding, we use confidence-based greedy decoding (Nie et al., 2024). For parallel sampling, we use confidence-aware parallel decoding with tuned confidence thresholds (Wu et al., 2025). All reported decoding throughput measurements use inference batch size 1 on a single H100 80GB GPU. For unconditional generation using models trained on OWT and LM1B, throughput is instead measured on four RTX A6000 48GB GPUs.

### L.1. Mathematical Reasoning

**Data**   For GSM8K, we distill a student model initialized from `Qwen/Qwen3-1.7B-Base` to support parallel diffusion decoding, following the adaptation approach of Arriola et al. (2025b). Distillation mitigates catastrophic forgetting by training the student on responses generated by a stronger teacher model, aligning the adaptation objective with the teacher's output distribution. Concretely, we construct the training set by prompting `Qwen/Qwen3-32B-Base` with each GSM8K training question and using the resulting responses as targets. We construct the validation set analogously by prompting `Qwen/Qwen3-32B-Base` with the GSM8K validation questions and using the resulting responses as validation targets.

The prompts are drawn from the `main` configuration of the GSM8K dataset (Cobbe et al., 2021), downloaded from `https://huggingface.co/datasets/openai/gsm8k`. We preprocess each input by prepending the instruction "`Please reason step by step, and put your final answer within $\boxed{}$.`". We preprocess each target by prepending "`Answer:   `" and formatting the final answer as "`$\boxed{}$.`". Inputs and targets are truncated to a maximum combined length of 1024 tokens.

**Tokenization and Hyperparameters**   We used the `Qwen/Qwen3-1.7B-Base` tokenizer. We use all the layers of the pretrained `Qwen/Qwen3-1.7B-Base` model, which has a hidden size of 2048 and intermediate hidden size of 6144. We train with batch size 1. The learning rate is linearly warmed up for 100 steps to a maximum of $1e^{-5}$ and decayed using a cosine schedule to half the peak value. Models are trained for up to 75k steps with early stopping based on the validation loss to select the model with the best validation perplexity.

**Evaluation**   Models generate responses with a maximum length of 1024 tokens. Generation is deterministic and uses the same Set Diffusion decoding procedure as above. We stop generations using a boxed-answer regular-expression criterion, with an additional repeating-token guard; runs using the chat-template format additionally stop on the `<|im_end|>` token. Inference is performed using the `lm-eval harness` library with the 'flexible match' criterion, and the same preprocessing is applied to question texts. Solutions are reverted to their original `### <Answer>` format.

In Figure 1, SW-SetDLM is trained with a maximum decoding window of 8 tokens, while block diffusion uses a block size of 4. At evaluation, both methods use a decoding block size of 4; SW-SetDLM supports sliding-window prediction and KV cache updates after every inference step, whereas block diffusion does not. We report zero-shot pass@1 accuracy and inference speed on an `H100 80GB` GPU over 200 samples, after a 50-sample warmup.

For parallel decoding, we tune the confidence threshold over the set $\{0.6, 0.65, 0.7, \ldots, 0.95, 0.99, 1.0\}$. For MDLM on GSM8K, we use a confidence threshold of 0.9, which we found to yield slightly better accuracy at higher throughput.

### L.2. Summarization

**Data**   For this task, we use the CNN/DailyMail dataset version 3.0 (Hermann et al., 2015; See et al., 2017) downloaded from https://huggingface.co/datasets/abisee/cnn_dailymail. The samples are preprocessed by adding a prefix to summaries: "`Summary:`   ". Inputs are truncated to a maximum length of 512 tokens and targets are truncated to a maximum length of 180 tokens, following the experimental setup of (Lin et al., 2023; Wu et al., 2023b).

**Tokenization and Hyperparameters**   We use the `Qwen/Qwen3-0.6B-Base` tokenizer. All models had a hidden size of 256, an intermediate hidden size of 768, and 28 transformer layers, corresponding to 80M parameters. We train with batch size 128. The learning rate is linearly warmed up for 1000 steps to a maximum of $3e^{-4}$. Models are trained for up to 500k steps with early stopping to select the model with the best validation perplexity.

**Evaluation**   Models generate up to 180 additional tokens. ROUGE scores (Lin, 2004) are computed using the `evaluate` library from HuggingFace. For all CNN/DailyMail models, we apply a repetition penalty of 1.2, require at least 50 generated tokens, and use an exponential length penalty starting at 80 tokens with decay factor 1.1. Decoding also uses an EOS/stop-string stopping criterion. Finally, for MDLM (Sahoo et al., 2024), we decode semi-autoregressively using a block size of 32.

### L.3. Likelihood Estimation and Unconditional Generation

**Data**   We train on OpenWebText (OWT; Gokaslan et al. (2019)) dataset and the One Billion Word dataset (LM1B; Chelba et al. (2014)). For OWT, we reserve the last 100K documents as a validation set. Following Arriola et al. (2025a), we concatenate documents and wrap them to the model context length without padding or truncation, and do not insert [BOS] or [EOS] tokens at the beginning and end of the context to enable arbitrary-length generation. We use a context length of 1024 for OWT and 128 for LM1B.

**Tokenization and Hyperparameters**   For OWT, we use the `gpt2` tokenizer (Radford et al., 2019); for LM1B, we use the `bert-base-uncased` tokenizer (Devlin et al., 2018). All models have 110M total parameters (excluding token embeddings). For models trained on OWT, we use a batch size of 512 and 300K total gradient updates. For models trained on LM1B, we use a batch size of 512 and 1M total gradient updates. The learning rate was linearly warmed up for 2500 steps to a peak of $3e^{-4}$ and held constant thereafter.

**Evaluation**   For likelihood evaluation, we use a single-sample Monte Carlo estimator over latent variables $t, \mathbf{z}_t, \sigma$. For unconditional generation, we sample 1K sequences from models trained on OWT and report MAUVE scores (Pillutla et al., 2021) against 1K samples from the OWT validation split. To reduce sensitivity to the particular validation subset, we fix each model's 1K generated samples and compute MAUVE against five fixed random subsets of 1K OWT validation examples, reporting the mean across subsets. We also report average sentence entropy for diversity. For categorical decoding, we sample directly from the model's predicted token distribution using multinomial sampling. Decoding throughput (`Tput`) is measured in tokens/sec on four RTX `A6000 48GB` GPUs. We report mean $\pm$ standard deviation over 200 post-warmup samples with maximum length $L \leq 1024$, gathered across ranks after 50 warmup examples per rank, using each model's configured sampling steps. For MAUVE, we use nucleus sampling with $p = 0.95$ and repetition penalty 1.05 for SetDLM and BD3LM, and $p = 0.90$ with no repetition penalty for AR. Generation stops after at least 128 tokens when the end-of-text confidence exceeds the stopping threshold for 4 consecutive windows of 128 tokens, or otherwise at $L = 1024$ tokens.

### L.4. Infilling

**Data**    We perform infilling evaluation using models trained on OWT, following (Guo & Ermon, 2025; Gong et al., 2025). For this task, we follow Guo & Ermon (2025) and test on 1871 short stories from the ROCStories dataset (Mostafazadeh et al., 2016) downloaded from Shen et al. (2023). Each story contains five sentences.

**Evaluation**    For each story, we mask either the middle sentence or the middle three sentences and have the model predict the missing span. ROUGE scores (Lin, 2004) are computed using the `evaluate` library from HuggingFace. We use a repetition-penalty logits processor during generation, with factor 1.2.

We record throughput on a single `H100 80GB` GPU.

## M. Perplexities on OpenWebText

Below, we provide the validation perplexities on OpenWebText. When block diffusion and set diffusion have comparable likelihoods, set diffusion achieves 22% faster decoding. MDLM is substantially slower because it lacks KV caching and recomputes the full 1024-token context at every denoising step.

*Table 8.* Validation perplexities (PPL; $\downarrow$) on OWT for SW-SetDLM trained for 157B tokens. [†] denotes numbers reported by Arriola et al. (2025a) which are trained for 524B tokens. Decoding throughput (`Tput`) is measured in tokens/sec on four RTX `A6000 48GB` GPUs. We report mean $\pm$ standard deviation over 200 post-warmup samples of maximum length $L \leq 1024$ using $T = L$, gathered across ranks after 50 warmup examples per rank.

|  | PPL ($\downarrow$) | `Tput` ($\uparrow$) |
|---|---|---|
| AR Transformer[†] | 17.54 | $96.7_{\pm 1.3}$ |
| *Diffusion* | | |
| SEDD[†] | $\leq 24.10$ | - |
| MDLM[†] | $\leq 22.98$ | $7.5_{\pm 0.01}$ |
| *AR+Diffusion Hybrid* | | |
| BD3LM[†] $S = 16$ | $\leq 23.34$ | $80.2_{\pm 0.6}$ |
| SW-SetDLM $S \leq 8$ | $\leq 22.42$ | $\mathbf{97.9_{\pm 1.6}}$ |

*Table 9.* Test perplexities (PPL; ↓) of models trained on 65B tokens on LM1B. Decoding throughput (Tput) is measured in tokens/sec on four RTX A6000 48GB GPUs. We report mean ± standard deviation over 200 post-warmup samples of length $L = 128$ using $T = L$, gathered across ranks after 50 warmup examples per rank. Best diffusion value is **bolded**. Models use 110M params., except those denoted by †: Transformer-X Base uses 0.5B params and D3PM (absorb) uses 70M params.

| Model | PPL (↓) | Tput (↑) |
|---|---|---|
| *Autoregressive* | | |
| Transformer-X Base[†] (Dai et al., 2019) | 23.5 | - |
| Transformer (Sahoo et al., 2024) | 22.83 | $120.1_{\pm 1.4}$ |
| *Diffusion* | | |
| D3PM[†] (absorb) (Austin et al., 2021) | $\leq 82.34$ | - |
| SEDD (Lou et al., 2023) | $\leq 32.68$ | - |
| MDLM (Sahoo et al., 2024) | $\leq 31.78$ | $\geq 37.7_{\pm 0.3}$ |
| UDLM (Schiff et al., 2025) | $\leq 36.71$ | - |
| Duo (Sahoo et al., 2025a) | $\leq 33.68$ | - |
| *AR+Diffusion Hybrid* | | |
| BD3LM $S = 16$ (Arriola et al., 2025a) | $\leq 30.60$ | $\geq 70.4_{\pm 0.8}$ |
| $S = 8$ | $\leq 28.95$ | $\geq 75.0_{\pm 0.8}$ |
| $S = 4$ | $\leq 28.23$ | $\geq 76.9_{\pm 1.2}$ |
| SW-SetDLM (*Ours*) $S \leq 32$ | $\leq 31.78$ | $\geq 92.8_{\pm 0.9}$ |
| $S \leq 16$ | $\leq 29.90$ | $\geq 96.7_{\pm 1.4}$ |
| $S \leq 8$ | $\leq \mathbf{27.83}$ | $\geq \mathbf{98.5_{\pm 1.6}}$ |

## N. Sample Generations

### N.1. Infilling

#### N.1.1. SW-SETDLM

**Infill 1 sentence**

One day I wanted to cook a nice dinner for my family. I decided to cook spaghetti. I made it with some olive oil, but after a while, I turned the stove on too high. The sauce burned and stuck to the bottom of the pan.

Ground truth: I boiled the noodles and put the sauce in a pot on the stove.

**Infill 3 sentences**

One day I wanted to cook a nice dinner for my family. It was not very good, but that's what happened with this one time recipe, because I just couldn't make it. The sauce burned and stuck to the bottom of the pan.

Ground truth: I decided to cook spaghetti. I boiled the noodles and put the sauce in a pot on the stove. I turned the stove on too high.

#### N.1.2. BD3LM

**Infill 1 sentence**

One day I wanted to cook a nice dinner for my family. I didn't want to do it again, but this time I decided to cook spaghetti. I turned the stove on too high. The sauce burned and stuck to the bottom of the pan.

Ground truth: I boiled the noodles and put the sauce in a pot on the stove.

**Infill 3 sentences**

One day I wanted to cook a nice dinner for my family.  One day I wanted to cook a
nice meal for my family.  I was in love with this dish, but it wasn't always easy.
The sauce burned and stuck to the bottom of the pan.

Ground truth:  I decided to cook spaghetti.  I boiled the noodles and put the
sauce in a pot on the stove.  I turned the stove on too high.

### N.1.3. MDLM

**Infill 1 sentence**

One day I wanted to cook a nice dinner for my family.  I decided to cook
spaghetti.  I cooked it in an oven with some pasta.  When I finished cooking, I
turned the stove on too high.  The sauce burned and stuck to the bottom of the
pan.

Ground truth:  I boiled the noodles and put the sauce in a pot on the stove.

**Infill 3 sentences**

One day I wanted to cook a nice dinner for my family.  I had no idea how to make
this recipe but it was pretty simple, so I took a large frying pan and put some
oil in it.  The sauce burned and stuck to the bottom of the pan.

Ground truth:  I decided to cook spaghetti.  I boiled the noodles and put the
sauce in a pot on the stove.  I turned the stove on too high.

### N.1.4. AR

**Infill 1 sentence**

One day I wanted to cook a nice dinner for my family.  I decided to cook
spaghetti.  It was the perfect meal, but it wasn't what we had I turned the stove
on too high.  The sauce burned and stuck to the bottom of the pan.

Ground truth:  I boiled the noodles and put the sauce in a pot on the stove.

**Infill 3 sentences**

One day I wanted to cook a nice dinner for my family.  I was in the middle of
cooking and had just finished eating some rice, which is what we were going to
eat together.  My husband The sauce burned and stuck to the bottom of the pan.

Ground truth:  I decided to cook spaghetti.  I boiled the noodles and put the
sauce in a pot on the stove.  I turned the stove on too high.

## N.2. Reasoning

### N.2.1. SW-SETDLM

For maximum training window size $S \leq 8$:

Question: Every day, Wendi feeds each of her chickens three cups of mixed chicken feed, containing seeds, mealworms and vegetables to help keep them healthy. She gives the chickens their feed in three separate meals. In the morning, she gives her flock of chickens 15 cups of feed. In the afternoon, she gives her chickens another 25 cups of feed. How many cups of feed does she need to give her chickens in the final meal of the day if the size of Wendi's flock is 20 chickens?

Answer:

```
We are given the following:

- Wendi has **20 chickens**.
- Each chicken gets **3 cups** of feed **every day**.
- The feed is divided into **three meals**:
  - **Morning**: 15 cups
  - **Afternoon**: 25 cups
  - **Final meal**: ?

---

### Step 1: Total feed needed per day

Each chicken gets 3 cups of feed per day, and there are 20 chickens:

20 chickens x 3 cups/chicken = 60 cups of feed

---

### Step 2: Total feed given in the first two meals

15 cups (morning) + 25 cups (afternoon) = 40 cups

---

### Step 3: Remaining feed for the final meal

60 cups (total) - 40 cups (first meals) = #### 20
```

For maximum training window size $S \leq 16$:

Question: Every day, Wendi feeds each of her chickens three cups of mixed chicken feed, containing seeds, mealworms and vegetables to help keep them healthy. She gives the chickens their feed in three separate meals. In the morning, she gives her flock of chickens 15 cups of feed. In the afternoon, she gives her chickens another 25 cups of feed. How many cups of feed does she need to give her chickens in the final meal of the day if the size of Wendi's flock is 20 chickens?

Answer:

```
We are told:

- Wendi has **20 chickens**.
- Each chicken gets **3 cups** of feed **every day**.
- The feed is divided into **3 meals**:
  - **Morning**: 15 cups
  - **Afternoon**: 25 cups
  - **Final meal**: ?

---

### Step 1: Total feed needed per day

Each of the 20 chickens gets 3 cups of feed:

20 chickens x 3 cups/chicken = 60 cups/day

---

### Step 2: Total feed given in the morning and afternoon

15 cups (morning) + 25 cups (afternoon) = 40 cups

---

### Step 3: Feed needed for the final meal

60 cups (total) - 40 cups (mor and afternoon) = #### 20
```

For maximum training window size $S \leq 32$:

Question: Every day, Wendi feeds each of her chickens three cups of mixed chicken feed, containing seeds, mealworms and vegetables to help keep them healthy. She gives the chickens their feed in three separate meals. In the morning, she gives her flock of chickens 15 cups of feed. In the afternoon, she gives her chickens another 25 cups of feed. How many cups of feed does she need to give her chickens in the final meal of the day if the size of Wendi's flock is 20 chickens?

Answer:

```
We are told that:

- Wendi has **20 chickens**.
- Each chicken gets **3 cups** of feed **3 different meals**.
- The feed is divided into **3 equal meals**.
- In the **morning**, she gives **15 cups**.
- In the **afternoon**, she gives **25 cups**.
- We are to find how many cups of feed are needed in the **final meal**.

---

### Step 1: Total feed given in the morning and afternoon

15 cups (morning) + 25 cups (afternoon) = 40 cups

---

### Step 2: Total feed needed for 20 chickens

Each chicken gets 3 cups, so:

20 chickens x 3 cups/chicken = 60 cups

---

### Step 3: Subtract the feed already given

60 cups (total) - 40 cups (used) = #### 20
```

### N.2.2. BD3LM

For block size $S = 4$:

Question: Every day, Wendi feeds each of her chickens three cups of mixed chicken feed, containing seeds, mealworms and vegetables to help keep them healthy. She gives the chickens their feed in three separate meals. In the morning, she gives her flock of chickens 15 cups of feed. In the afternoon, she gives her chickens another 25 cups of feed. How many cups of feed does she need to give her chickens in the final meal of the day if the size of Wendi's flock is 20 chickens?

Answer:

```
Let's break this down step by step:
---
### **Step 1: Total feed needed per day**
Wendi feeds **3 cups of feed per chicken per day**.
She has **20 chickens**, so:

20 chickens x 3 cups per chicken = 60 cups per day

---
### **Step 2: Feed already given in the morning**
She gives **15 cups** in the morning.
---
### **Step 3: Feed already given in the afternoon**
She gives **25 cups** in the afternoon.
---
### **Step 4: Total feed given so far**

15 cups (morning) + 25 cups (afternoon) = 40 cups

---
### **Step 5: Remaining feed for the final meal**

60 cups (total) - 40 cups (given) = #### 20
```

For block size $S = 8$:

Question: Every day, Wendi feeds each of her chickens three cups of mixed chicken feed, containing seeds, mealworms and vegetables to help keep them healthy. She gives the chickens their feed in three separate meals. In the morning, she gives her flock of chickens 15 cups of feed. In the afternoon, she gives her chickens another 25 cups of feed. How many cups of feed does she need to give her chickens in the final meal of the day if the size of Wendi's flock is 20 chickens?

Answer:

```
We are told:

- Wendi has **20 chickens**.
- Each chicken receives **3 cups of feed per day**.
- The feed is given in **three meals**:
  - **Morning**: 15 cups
  - **Afternoon**: 25 cups
  - **Final meal**: ?

---

### Step 1: Total feed needed for 20 chickens

Each chicken gets 3 cups of feed per day:

20  chickens x 3 cups per chicken = 60 cups per day

---

### Step 2: Total feed given in the morning and afternoon

15 cups + 25 cups = 40 cups

---

### Step 3: Total feed needed for the final meal

60 cups - 40 cups = #### 20
```

For block size $S = 16$:

Question: Every day, Wendi feeds each of her chickens three cups of mixed chicken feed, containing seeds, mealworms and vegetables to help keep them healthy. She gives the chickens their feed in three separate meals. In the morning, she gives her flock of chickens 15 cups of feed. In the afternoon, she gives her chickens another 25 cups of feed. How many cups of feed does she need to give her chickens in the final meal of the day if the size of Wendi's flock is 20 chickens?

Answer:

```
We are told:

- Wendi feeds **3 cups of feed per chicken**.
- She has **20 chickens**.
- She gives the feed in **three meals**:
  - **Morning**: 15 cups
  - **Afternoon**: 25 cups
  - **Final meal**: We need to find how many cups of feed she needs to give the
    **chickens in the final meal**.

---

### Step 1: Total feed needed for 20 chickens

Each chicken gets 3 cups of feed so:

20 chickens x 3 cups/chicken = 60 cups

---

### Step 2: Total feed given in first two meals

15 cups (mor) + 25 cups (afternoon) = 40 cups

---

### Step 3: Total feed needed in the final meal

60 cups (total) - 40 cups (first two meals) = #### 20
```

### N.2.3. AR

Question: Every day, Wendi feeds each of her chickens three cups of mixed chicken feed, containing seeds, mealworms and vegetables to help keep them healthy. She gives the chickens their feed in three separate meals. In the morning, she gives her flock of chickens 15 cups of feed. In the afternoon, she gives her chickens another 25 cups of feed. How many cups of feed does she need to give her chickens in the final meal of the day if the size of Wendi's flock is 20 chickens?

Answer:

```
Let's break this down step by step:

### Step 1: Total feed given in the morning
Wendi gives **15 cups** of feed in the morning.

### Step 2: Total feed given in the afternoon
She gives **25 cups** of feed in the afternoon.

### Step 3: Total feed given so far

15 cups + 25 cups = 40 cups

### Step 4: Total feed needed for the day
Wendi has **20 chickens**, and each chicken gets **3 cups** of feed per day:

20 chickens x 3 cups/chicken = 60 cups

### Step 5: Feed needed in the final meal

60 cups - 40 cups = #### 20
```

### N.2.4. MDLM

Question: Every day, Wendi feeds each of her chickens three cups of mixed chicken feed, containing seeds, mealworms and vegetables to help keep them healthy. She gives the chickens their feed in three separate meals. In the morning, she gives her flock of chickens 15 cups of feed. In the afternoon, she gives her chickens another 25 cups of feed. How many cups of feed does she need to give her chickens in the final meal of the day if the size of Wendi's flock is 20 chickens?

Answer:

Let's break this down step by step.

---

### **Step 1: Total eggs per day**

Janet's ducks lay **16 eggs per day**.

---

### **Step 2: Eggs eaten for breakfast**

Janet she eats **3 eggs for breakfast every morning**.

So, she eats **3 eggs per day**.

---

### **Step 3: Eggs eaten for muffins**

Janet she bakes **4 muffins for her friends every day with four**.

So, she bakes **4 eggs per day**.

---

### **Step 4: Total eggs eaten**

Total eggs eaten = Breakfastfast + Muffins + Remaining eggs

Total eggs eaten = 3 + 4 + 4 = 11 eggs eaten

---

### **Step 5: Remaining eggs**

Janet's ducks lay **16 eggs per day**, and she eats **11 eggs eaten**.

Remaining eggs = 16 – 11 = 5 eggs remaining

---

### **Step 6: Selling the remaining eggs**

She sells the **remaining eggs** at the farmers' market for **\$2 per duck egg**.

So, she sells **\$10 per day**.

---

### **Step 7: Total money made**

She sells **\$10 per day**, and she sells **5 remaining eggs**.

Total money made = 10 \times 5 = 50 dollars

...

