# OpenReview forum: "Set Diffusion: Interpolating Token Orderings between Autoregression and Diffusion for Fast and Flexible Decoding"
_ICML.cc/2026/Conference — ICML 2026 regular_

### Official Review · Reviewer_zBzR · 2026-02-14

**Soundness:** 2
**Presentation:** 3
**Significance:** 2
**Originality:** 2
**Overall Recommendation:** 3
**Confidence:** 4

**Summary:**

This work presents SetDLM, an interpolation between AR and MDLM from the perspective of decoding order AR-ness. With an order-informed noise scheduler and its corresponding diffusion training objective, SetDLM could control the left-to-right bias in decoding and obtain a tighter likelihood bound in training. Based on the any-order AR reparameterization, SetDLM could support full KV caching. Experiments show better performance of SetDLM over another important interpolation method of block diffusion.

**Compliance With Llm Reviewing Policy:**

Affirmed.

**Key Questions For Authors:**

Please see the weaknesses section.

**Limitations:**

yes

**Strengths And Weaknesses:**

## Strengths

- The order-based interpolation between AR and DLM has several advantages over the block diffusion method, e.g. tighter NELBO and full KV caching.
- Experiments of mathematical reasoning, summarization, and unconditional generation, demonstrates the superiority of SetDLM over the block diffusion baseline.
- The writing and presentation of the method is clear.

## Weaknesses

- The novelty of the proposed method is questionable. The two core contributions in this work are non-uniform noise scheduler and causal diffusion architecture, which are not completely novel ideas. For example, the position-aware noise scheduler and training objective in [1] are very similar to the corresponding ones in this work. Besides, there are also multiple causal DLMs that support full KV caching have been proposed [2, 3]. Therefore, this paper needs more comparisons and discussions on those related works, especially [1].
- As many works are focusing on learning the decoding order (or noise scheduler) of DLMs [4, 5, 6], I have concerns about whether the scheduler only based on left-to-right bias could surpass those learning-based approaches.
- Although the model is scaled up to 1.7B parameters, this paper only includes GSM8K evaluations of the 1.7B model, I think more downstream task evaluations are missing, for example, the likelihood-based commonsense reasoning tasks [7] for 1.7B model or zero-shot language modeling perplexities for small models, which will be helpful to more thoroughly demonstrate the model's performance.
- The reason why SetDLM gains better performance than block diffusion seems unclear, more theoretical analysis of the connection and advantage over block diffusion could be beneficial.

[1] Garg P, Kohli B, Sarawagi S. Masked Diffusion Models are Secretly Learned-Order Autoregressive Models[J]. arXiv preprint arXiv:2511.19152, 2025.

[2] Sahoo S S, Yang Z, Akhauri Y, et al. Esoteric Language Models[J]. arXiv preprint arXiv:2506.01928, 2025.

[3] Li J N, Guan J, Wu W, et al. ReFusion: A Diffusion Large Language Model with Parallel Autoregressive Decoding[J]. arXiv preprint arXiv:2512.13586, 2025.

[4] Peng F Z, Bezemek Z, Patel S, et al. Path planning for masked diffusion model sampling[J]. arXiv preprint arXiv:2502.03540, 2025.

[5] Peng F Z, Bezemek Z, Rector-Brooks J, et al. Planner Aware Path Learning in Diffusion Language Models Training[J]. arXiv preprint arXiv:2509.23405, 2025.

[6] Hong C, An S, Kim M S, et al. Improving Discrete Diffusion Unmasking Policies Beyond Explicit Reference Policies[J]. arXiv preprint arXiv:2510.05725, 2025.

[7] Nie S, Zhu F, Du C, et al. Scaling up masked diffusion models on text[J]. arXiv preprint arXiv:2410.18514, 2024.

---

> ### Author Rebuttal · Authors · 2026-03-31
>
> Thank you for the thorough review. Below we address the concerns.
> # W1: Evaluation on commonsense reasoning
> We provide new commonsense results using 110M models trained on OWT as they reflect general-purpose language modeling. A general-purpose 1.7B checkpoint would be ideal, but our available 1.7B models were finetuned on GSM8K and would confound commonsense evaluation.
>
>
> **Zero-shot commonsense reasoning (110M, OWT). Best diffusion bolded.**
>
> |Model|HellaSwag|PIQA|SIQA|Avg↑|
> |-|-|-|-|-|
> |AR|31.70|62.30|39.25|44.42|
> |MDLM[8]|26.80|51.20|34.85|37.62|
> |BD3LM S=16[7]|26.80|51.03|35.36|37.73|
> |BD3LM S=8|26.95|53.21|34.14|38.10|
> |BD3LM S=4|27.69|52.12|33.52|37.78|
> |SetDLM S≤32 (Ours)|29.51|58.71|38.84|42.35|
> |SetDLM S≤16 (Ours)|29.77|59.68|38.18|42.55|
> |SetDLM S≤8 (Ours)|**30.33**|**61.37**|**39.10**|**43.60**|
>
> Following [9], we restrict to benchmarks scoring above chance at this scale; harder benchmarks (ARC-Challenge, WinoGrande, MMLU) yield near-random performance for both AR and diffusion models.
> # W2: Comparison to non-uniform noise schedulers [1, 2, 3]
> We compare against [1,2,3] with new 1.7B-scale GSM8K results. Garg et al. [1] observe that MDLMs implicitly learn position-dependent unmasking orders at inference, but propose no new noise process or training objective. SetDLM makes ordering explicit at training time via a per-token noise process (Def. 4.2) that provably tightens the NELBO (Prop. 4.1) while enabling KV caching and variable-length generation:
>
> ||Garg et al. [1]|EsoLMs [2]|ReFusion [3]|SetDLM (Ours)|
> |-|-|-|-|-|
> |Tunable L2R bias|-|-|-|✓|
> |Full KV caching|-|✓|✓|✓|
> |All-token training|-|-|-|✓|
> 1. **Noise schedule.** SetDLM interpolates between AR and diffusion, recovering both as special cases. [1] uses inference-time reordering only; [2] splits into disjoint AR/diffusion stages; [3] decodes AR within fixed blocks w/o likelihood estimation.
> 2. **KV caching + variable-length gen.** [1] does not address this; [2,3] support KV caching but not variable-length generation.
> 3. **All-token training.** Unlike [1,2,3] which only compute gradients for masked tokens, our training reduces gradient variance by ~3× (Tab. 6).
>
> We add a 1.7B-scale GSM8K comparison under sequential sampling:
>
> ||GSM8K Acc ↑|
> |-|-|
> |EsoLMs [2] α₀ = 0.0625|17.21|
> |ReFusion [3]|57.70|
> |SetDLM S≤32 (Ours)|61.94|
>
> We report EsoLMs’ best α₀ on GSM8K across those in [2]. EsoLMs’ α₀ does not effectively tune L2R bias, degrading quality as tokens sampled far from context compound decoding errors. ReFusion [3] degrades at higher parallelism for the same reason: AR decoding within blocks compounds errors when blocks are generated in parallel.
>
> # W3: Generalization to learned orders [4], [5], [6]
> The per-token noise parameterization is compatible with learned orderings; extending to data-dependent $q(\sigma|\mathbf{x})$ is a natural future direction. Unlike [4,5,6] which retrofit ordering via inference-time planners, loss reweighting, or RL, SetDLM encodes ordering directly in the noise schedule (Def 4.2) and optimizes an unbiased NELBO (Eq 8):
>
> - **P2 [4]** applies planning only at inference, creating a train-test mismatch; SetDLM is order-aware at both training and inference.
> - **PAPL [5]** optimizes a biased NELBO reweighting losses by model confidence; SetDLM uses an unbiased NELBO.
> - **UPO [6]** learns a separate RL policy with rollout overhead, lacking KV caching or variable-length generation; SetDLM optimizes likelihood directly with both.
>
> ||P2 [4]|PAPL [5]|UPO [6]|SetDLM (Ours)|
> |-|-|-|-|-|
> |Learned orders at training|-|-|✓|✓|
> |Full KV caching|-|-|-|✓|
> |All-token training|-|-|-|✓|
> |Variable-length gen|-|-|-|✓|
>
> We compare to PAPL [5] on GSM8K at the 1.7B scale (sequential decoding):
>
> ||GSM8K Acc ↑|
> |-|-|
> |PAPL [5]|5.23|
> |SetDLM S≤32 (Ours)|**61.94**|
>
> [5] does not support variable-length generation, producing degenerate run-on responses on GSM8K.
> # W4: Theoretical advantage over Block Diffusion [7]
> SetDLM strictly generalizes [7]; setting $w=S/L, k=1$ recovers it as a special case. [7] uses non-overlapping blocks (one must complete before the next); SetDLM allows overlapping intervals for sliding-window decoding and arbitrary infilling. At 1.7B, SetDLM outperforms Block Diffusion on GSM8K (Fig 1, Tab 1) and achieves higher ROUGE with ≤25% faster infilling (Tab 3).
>
> **References**
>
> [1] Garg et al. “Masked Diffusion Models are Secretly Learned-Order Autoregressive Models” 2025
>
> [2] Sahoo et al. “Esoteric Language Models” 2025
>
> [3] Li et al. “ReFusion” 2026
>
> [4] Peng et al. “Path Planning for Masked Diffusion Model Sampling” 2025
>
> [5] Peng et al. “Planner Aware Path Learning in Diffusion Language Models Training” 2026
>
> [6] Hong et al. “Improving Discrete Diffusion Unmasking Policies” 2026
>
> [7] Arriola et al. “Interpolating autoregressive and discrete denoising diffusion language models” 2025
>
> [8] Sahoo et al. “Simple and effective masked diffusion language models” 2024
>
> [9] Nie et al. “Scaling up Masked Diffusion Models on Text” 2025

---

> > ### Author Rebuttal · Reviewer_zBzR · 2026-04-01
> >
> > Thanks for your response.
> >
> > W1: about the experiments, not fully addressed.
> >
> > 1.1: "... finetuned on GSM8K and would confound commonsense evaluation", why not evaluating the 1.7B model before fine-tuning?
> >
> > 1.2: the zero-shot language modeling perplexities on other datasets are still not reported. [1, 2]
> >
> > W2 & W3 & W4: addressed. More discussions on other methods have been supplemented.
> >
> > [1] Subham Sahoo, Marianne Arriola, Yair Schiff, Aaron Gokaslan, Edgar Marroquin, Justin Chiu, Alexander Rush, and Volodymyr Kuleshov.Simple and effective masked diffusion language models.Advances in Neural Information Processing Systems, 37:130136–130184, 2024.
> >
> > [2] Jingyang Ou, Shen Nie, Kaiwen Xue, Fengqi Zhu, Jiacheng Sun, Zhenguo Li, and Chongxuan Li.Your absorbing discrete diffusion secretly models the conditional distributions of clean data.In The Thirteenth International Conference on Learning Representations, 2025.

---

> > > ### Author Response · Authors · 2026-04-03
> > >
> > > Thank you for the follow-up! Since W2-W4 are resolved, we address only the remaining experimental point below.
> > >
> > > # W1.1: Why not evaluate the 1.7B model before GSM8K fine-tuning?
> > >
> > > Our 1.7B models are Qwen-initialized checkpoints adapted specifically for GSM8K: evaluating them "before fine-tuning" would evaluate the AR Qwen backbone, not a trained 1.7B diffusion model. We do not have a general-purpose 1.7B SetDLM checkpoint because training one from scratch was beyond our academic compute budget.
> > >
> > > This experimental setup of diffusion fine-tuning from an AR backbone for larger-scale experiments is consistent with DiffuGPT [1], ReFusion [2], and E2D2 [3], and is a recognized approach in the diffusion LM literature for evaluating at scale without prohibitive pretraining cost. Our general-purpose evaluations are therefore reported at 110M, where all models are trained from scratch.
> > >
> > > # W1.2: New zero-shot perplexities
> > >
> > > We follow the reviewer's valuable guidance and provide new zero-shot perplexities (↓) on benchmark datasets from [6] for 110M models trained from scratch for 300K steps on OWT following [4,5], where the best diffusion value is bolded. We compare throughput-matched checkpoints: SetDLM S≤8 and BD3LM S=16 (the fastest BD3LM matching SetDLM's throughput), decoding at ~81 and ~76 tok/s on A100 over 100 samples.
> > >
> > > SetDLM achieves the **best diffusion perplexity on 4 of 7 benchmarks while decoding faster than BD3LM**, and achieves **lower perplexity than the AR transformer on 4 of 7 benchmarks**, demonstrating superior out-of-domain generalization:
> > >
> > > |Dataset|SetDLM S≤8 (Ours)|BD3LM S=16|MDLM|\||AR|
> > > |-|-:|-:|-:|:-:|-:|
> > > | Throughput (tok/s; ↑)|**$\mathbf{80.85_{±1.67}}$**|$76.37_{±3.28}$|$55.11_{± 3.67}$|\||$71.49_{±0.35}$|
> > > |PTB|101.69|**97.67**|102.27|\||89.06|
> > > |WikiText|**33.96**|36.18|38.53|\||32.97|
> > > |LM1B|**65.83**|67.44|71.90|\||50.91|
> > > |LAMBADA |**50.52**|56.13|89.86|\||85.52|
> > > |AG News|**76.05** |77.35|136.73|\||128.29|
> > > |PubMed|51.78|**50.04**|55.20|\||74.35|
> > > |Arxiv|60.76|**57.07**|61.86|\||89.34|
> > >
> > > Throughput is measured under confidence-based parallel sampling [7], where tokens with prediction confidence ≥95% are accepted in parallel. We agree that evaluating general-purpose 1.7B SetDLM checkpoints would have been ideal, but training such a model is beyond our academic compute budget. We hope the table above fully resolves the remaining experimental question.
> > >
> > > # Summary of evaluation breadth
> > >
> > > With these additions, SetDLM has been evaluated across six evaluation axes: perplexity (Tables 5, 9, and the table above), 0-shot commonsense reasoning (rebuttal), mathematical reasoning (Table 1), summarization (Table 2), infilling (Table 3), and unconditional generation (Table 4).
> > >
> > > **With W2–W4 resolved and the requested zero-shot perplexities provided for W1, we believe all four original concerns have been addressed. We respectfully ask the reviewer to consider whether the updated evidence supports reconsideration of the score. Many thanks for your valuable guidance and constructive engagement throughout!**
> > >
> > > [1] Gong et al. "Scaling Diffusion Language Models via Adaptation from Autoregressive Models" 2025
> > >
> > > [2] Li et al. “ReFusion” 2026
> > >
> > > [3] Arriola et al. “Encoder-Decoder Diffusion Language Models” 2025
> > >
> > > [4] Sahoo et al. "Simple and Effective Masked Diffusion Language Models" 2024
> > >
> > > [5] Arriola et al. “Interpolating autoregressive and discrete denoising diffusion language models” 2025
> > >
> > > [6] Radford et al. "Language Models are Unsupervised Multitask Learners" 2019
> > >
> > > [7] Wu et al. “Fast-dLLM: Training-free Acceleration of Diffusion LLM by Enabling KV Cache and Parallel Decoding” 2025

---

### Official Review · Reviewer_EMsh · 2026-03-03

**Soundness:** 2
**Presentation:** 3
**Significance:** 3
**Originality:** 2
**Overall Recommendation:** 3
**Confidence:** 3

**Summary:**

This paper proposes Set Diffusion Language Models (SetDLM), a new class of discrete diffusion models designed to bridge the gap between autoregressive (AR) and diffusion-based decoding. The authors argue that interpolating generation orderings is more effective than the fixed block segmentation used in prior works like Block Diffusion. The approach introduces an order-informed noise process that biases decoding toward left-to-right generation while maintaining flexibility for any-order tasks like infilling. Key technical contributions include a causal diffusion architecture that enables full Key-Value (KV) caching and a token-efficient training algorithm that computes gradients for all tokens in a batch simultaneously. Experimental results across mathematical reasoning (GSM8K), summarization (CNN/DailyMail), and language modeling (OWT, LM1B) demonstrate that SetDLM achieves a superior speed-accuracy tradeoff compared to existing discrete diffusion baselines.

**Compliance With Llm Reviewing Policy:**

Affirmed.

**Ethical Review Concerns:**

I have read the authors' rebuttal carefully and appreciate the clarifications regarding computational constraints, noise schedule parameters, and the supplementary training efficiency data. However, after re-evaluating the manuscript in light of these responses, I will maintain my score of Weak Reject. My primary reservations regarding the empirical tradeoffs and the absolute performance gap remain unresolved.

**Final Justification:**

I have read the authors' rebuttal carefully and appreciate the clarifications regarding computational constraints, noise schedule parameters, and the supplementary training efficiency data. However, after re-evaluating the manuscript in light of these responses, I will maintain my score of Weak Reject. My primary reservations regarding the empirical tradeoffs and the absolute performance gap remain unresolved.

**Key Questions For Authors:**

1. In Figure 1, there is a clear drop in accuracy as tokens/sec increase. Is there a theoretical "limit" to how much accuracy is sacrificed for a given $\overline{C}$, or do you believe further architecture optimizations could push the SetDLM frontier closer to the AR point?

2. Most experiments use 110M to 1.7B parameter models. Have you observed any "emergent" behaviors or changes in the noise schedule's effectiveness when scaling to larger architectures?

3. Table 6 shows significantly lower gradient variance. Does this efficiency allow for a reduction in the total number of training tokens required to reach a specific perplexity compared to MDLM?

4. The paper mentions "any-order decoding up to 25 positions ahead". How was this specific lookahead value chosen, and does increasing it further significantly degrade the quality of the generated text?

**Limitations:**

yes

**Strengths And Weaknesses:**

Strengths:
- The Paper introduces a tunable ordering distribution that recovers masked diffusion under specific per-token noise schedules. This provides a more principled and flexible mathematical framework for interpolating between AR and diffusion than rigid block structures. It allows for a smooth tradeoff between sample quality and parallelization by adjusting the decoding window size $w$.
- The proposed causal architecture and the sampling algorithm (Algorithm 1) demonstrate full KV caching capabilities. This addresses a major limitation of standard diffusion models which typically require bidirectional context and fixed-length vectors. As shown in Table 5, SetDLM achieves 141.8 tokens/sec, outperforming MDLM's 121.6 tokens/sec on LM1B.
- Table 6 reports that the "efficient training" method reduces gradient variance by approximately 3x (from $7.50 \times 10^{-2}$ to $2.73 \times 10^{-2}$). Moving beyond objectives that only update masked tokens improves the utilization of training data. This leads to better likelihood estimation and faster convergence compared to the MDLM baseline.
- Figure 1 and Table 1 show that SetDLM (S≤8) achieves 66.41% accuracy on GSM8K, outperforming Block Diffusion (S=4) at 63.53% with higher throughput. It demonstrates that the theoretical benefits of staggered noise schedules translate to practical reasoning gains. The model shows state-of-the-art speed-quality tradeoffs among discrete diffusion models.

Weaknesses:
- The noise schedule depends on both the unmasking interval width $w$ and the scaling exponent $k$ (Section C). The "ideal schedule" is stated to depend on specific user preferences and hardware capabilities, which may limit out-of-the-box usability. Finding the optimal $w$ and $k$ to satisfy the expected inference budget $\overline{C}$ while maintaining lookahead flexibility adds significant tuning overhead. Provide a sensitivity analysis or a grid search visualization of how varying $k$ independently of $w$ affects performance on a validation set.
- Table 1 shows the AR Transformer at 80.21% accuracy on GSM8K, while the best SetDLM is at 66.41%. While the gap is narrowing, the tradeoff between parallelization and accuracy remains a hurdle for diffusion models in high-stakes reasoning tasks. The "Improved Tradeoff" indicated in Figure 1 still places the diffusion models significantly below the AR star on the y-axis. Evaluate if scaling the model size (e.g., to 7B or 14B) reduces this specific accuracy gap compared to an AR model of equivalent size.

---

> ### Author Rebuttal · Authors · 2026-03-31
>
> Thank you for the constructive feedback. Below we address the comments raised.
> # W1: Scaling to 7B/14B
> Scaling to 7B/14B is beyond our academic compute resources; we hope the reviewer will evaluate our contributions at the scales we were able to study. In our evaluations at the 1.7B and 110M scales (15x model-size range) across mathematical reasoning, infilling, likelihood estimation, summarization, and commonsense reasoning (see our response to reviewer zBzR W1) SetDLM demonstrates consistent improvements over diffusion baselines. All-token training reduces gradient variance by ~3× (Tab. 6), and full KV caching accelerates inference at both scales.
> # W2: Sensitivity of noise schedule parameters
> We clarify that the user need only tune scalar inference prediction budget $\bar{C}$. The decoding width $w$ should be maximized to permit any-order decoding, for which we use a default $w = 4\bar{C}/L$ because it is the most robust setting across operating points and naturally corresponds to a maximum lookahead that is twice that of block diffusion (see below), and compute $k$ deterministically as $k = \frac{\bar{C}}{Lw − \bar{C}}$ (App C2). Thus, normal use does not require a 2D grid search over $(w,k)$.
>
> In our experiments comparing to block diffusion, we further clarify that our default maximum lookahead (from choice of $w$) is 2S (2x that of block diffusion), not “25” as stated in the review.
>
> We fix $\bar{C}$ and record GSM8K accuracy from training 1.7B models with varying $w$:
>
> |w|2x parallel|3x parallel|4x parallel|
> |-|-|-|-|
> |$2\bar{C}/L$|60.65|55.04|41.24|
> |$4\bar{C}/L$|61.87|**58.38**|**49.51**|
> |$8\bar{C}/L$|**63.46**|58.23|49.28|
>
> Our default $w=4\bar{C}/L$ is the most robust setting across the full parallelism range.
>
> The reviewer also asks about varying w independently of k. By construction, w and k are coupled through the max lookahead (Eqs. 35, 40): varying one while fixing the other changes the expected inference budget $\bar{C}$. We vary w while fixing $k=1$ at 1.7B and report GSM8K accuracy:
>
> |w|Parallelism factor (↑)| GSM8K acc. (↑)|
> |-|-|-|
> |4/L|1.00 ± 0.00|65.81|
> |4/L|1.94 ± 0.23|66.19|
> |4/L|2.20 ± 0.26|63.00|
> |16/L|1.00 ± 0.00|60.58|
> |16/L|2.38 ± 0.40|59.06|
> |16/L|2.88 ± 0.51|55.04|
>
> Larger decoding windows degrade GSM8K accuracy, as tokens sampled further from available context introduce compounding decoding errors.
> # W3: Speed-quality tradeoff versus AR on GSM8K
> SetDLM advances the diffusion speed-quality frontier; it does not claim to beat AR. **In Fig. 1, SetDLM S≤8 achieves 66.41% accuracy at 45.25 tok/s, compared to Block Diffusion S=4 at 63.53% at 40.51 tok/s [1].** The AR transformer achieves 80.21%; the contribution is the improved diffusion Pareto frontier.
>
> # Addressing Questions
> > Observed scaling trends
>
> We do not claim a scaling law from the current evidence. SetDLM’s advantage over diffusion baselines is larger in our 1.7B setting than in our 110M setting. On LM1B at 110M, SetDLM improves over MDLM from 31.78 to 27.85 PPL and over the best BD3LM setting from 28.23 to 27.85 (Tab. 5); on GSM8K at 1.7B, SetDLM improves over $\bar{C}$-matched BD3LM settings by +11.45, +6.97, and +2.88 pass@1 (Fig. 1, Tab. 1). Because these are different tasks, we present this only as cross-scale consistency, not as evidence that the AR gap closes with scale.
>
> > Theoretical speed-quality limit
>
> Prop. 4.1 does not prove a universal accuracy-vs-throughput lower bound for finite-capacity models. It identifies one source of the gap: wider schedules loosen the variational bound, whereas $w \to 1/L$ removes that schedule-induced gap and recovers the AR NLL exactly. Better orderings (e.g., data-dependent) could tighten the NELBO, or denoisers that more closely approximate the posterior would move the empirical frontier closer to AR.
>
> > Gradient variance and training efficiency
>
> We find that MDLM needs to train on ~30% more tokens than SetDLM to reach the same LM1B perplexity (under the same linear noise schedule applied on all tokens), consistent with our all-token objective yielding ~3× lower gradient variance and lower perplexity at matched training budgets on LM1B.
>
> When we tune the schedule by tending to AR generation for lower $\bar{C}$, we obtain up to 70% fewer training tokens than MDLM.
>
> |Model|# training tokens to PPL≤40|% training token reduction from MDLM to PPL ≤ 40|
> |-|-|-|
> |MDLM [2]|6.6B|-|
> |SetDLM S≤1024|5.2B|21%|
> |SetDLM S≤16|3.9B|41%|
> |SetDLM S≤8|2.6B|61%|
> |SetDLM S≤4|2.0B|70%|
>
> **References**
>
> [1] Arriola et al. “Interpolating autoregressive and discrete denoising diffusion language models” 2025.
>
> [2] Sahoo et al. “Simple and effective masked diffusion language models” 2024.
>
> [3] Wu et al. “Fast-dLLM: Training-free Acceleration of Diffusion LLM by Enabling KV Cache and Parallel Decoding” 2025.

---

> > ### Author Rebuttal · Reviewer_EMsh · 2026-04-02
> >
> > I have read the authors' rebuttal carefully and appreciate the clarifications regarding computational constraints, noise schedule parameters, and the supplementary training efficiency data. However, after re-evaluating the manuscript in light of these responses, I will maintain my score of Weak Reject. My primary reservations regarding the empirical tradeoffs and the absolute performance gap remain unresolved.
> > Regarding the noise schedule sensitivity, I thank the authors for clarifying that users primarily tune the expected inference prediction budget $\overline{C}$ and for providing the ablation tables that vary $w$ and $k$. Unfortunately, these additional results actually reinforce my initial concerns. The tables clearly show a steep drop in GSM8K accuracy as the parallelism factor increases, plummeting from 65.81% down to 55.04%. This indicates that the smooth interpolation severely penalizes the model's reasoning capabilities. As the authors correctly point out, larger decoding windows introduce compounding decoding errors. This inherent fragility when handling long dependencies or complex reasoning sequences remains a fundamental flaw of the proposed approach.

---

> > > ### Author Response · Authors · 2026-04-03
> > >
> > > Thank you for the careful re-reading and for specifying the two remaining concerns. We address each directly below.
> > >
> > > # W3.1: On the accuracy-parallelism tradeoff as "inherent fragility"
> > >
> > > We clarify a critical point: **the degradation the reviewer identifies is not specific to SetDLM**, it is the inherent cost of parallel decoding in *any* discrete diffusion method, where each token is predicted conditioned on fewer revealed tokens. The 65.81%→55.04% drop the reviewer cites spans an aggressive **~3× increase** in sampling parallelism, precisely the regime where any diffusion decoder pays the largest quality cost for speed.
> > >
> > > The relevant question is: **does SetDLM handle this tradeoff better than prior work?** The answer is yes. When both models decode the same average tokens per step, SetDLM **simultaneously achieves higher accuracy and higher throughput** than Block Diffusion (BD3LM, ICLR 2025 Oral [1]):
> > >
> > > |Parallelism|BD3LM S=4 Acc|Speed (tok/s)|SetDLM S≤8 Acc|Speed (tok/s)| SetDLM acc gain |
> > > |-|-|-|-|-|-|
> > > |1.0×|63.53|40.5|**66.41**|**45.3**|**+2.88**|
> > > |~1.8×|63.31|61.1|**67.10**|**80.8**|**+3.79**|
> > > |~2.2×|60.05|69.3|**63.68**|**96.8**| **+3.63**|
> > > |~2.5×|54.97|75.3|**58.23**|**107.2**|**+3.26**|
> > >
> > > BD3LM drops from 63.53% to 54.97% over this range, consistently worse than SetDLM. SetDLM's Pareto frontier **strictly dominates** the prior state of the art at every operating point. This makes SetDLM's tradeoff curve stronger, not more fragile, than existing diffusion methods. This advantage extends beyond Block Diffusion: In the 1.7B sequential decoding, SetDLM S≤32 achieves 61.94 on GSM8K vs. 57.70 for ReFusion [2] and 17.21 for EsoLMs [3].
> > >
> > > We hope our work is judged on its central claim that order-informed diffusion improves the Pareto frontier over prior diffusion LMs.
> > >
> > > # W3.2: On the "absolute performance gap" between AR and Set Diffusion
> > >
> > > We agree that a quality gap to AR exists, and we want to be precise: **SetDLM does not claim to close this gap.** The paper's contribution is advancing the diffusion LM Pareto frontier relative to prior diffusion methods, where the gap is even larger. That said, diffusion LMs offer concrete structural advantages over AR that motivate this research direction:
> > >
> > > ### **1. Better generalization to new domains**
> > >
> > > Diffusion models must predict tokens from partial, unordered context rather than a complete left prefix as in AR, which acts as implicit regularization against memorizing domain-specific patterns. This out-of-domain generalization advantage has also been observed by [1,3,4].
> > >
> > > We provide new zero-shot perplexities (↓) on held-out benchmarks [5] for 110M models trained on OWT following [1,3,4]. We compare throughput-matched checkpoints on 1 A100 over 100 samples: SetDLM S≤8 and BD3LM S=16 (the fastest BD3LM matching SetDLM's throughput).
> > >
> > > SetDLM achieves the **best diffusion perplexity on 4 of 7 benchmarks while decoding faster than BD3LM**, and achieves **lower perplexity than the AR transformer on 4 of 7 benchmarks**, demonstrating superior out-of-domain generalization at higher throughput:
> > >
> > > |Dataset|SetDLM S≤8 (Ours)|BD3LM S=16|AR|MDLM|
> > > |-|-:|-:|-:|-:|
> > > | Throughput (tok/s)|80.85±1.67|76.37±3.28|71.49±0.35|55.11±3.67|
> > > |PTB|101.69|97.67|89.06|102.27|
> > > |WikiText|33.96|36.18|32.97| 38.53|
> > > |LM1B|65.83|67.44|50.91|71.90|
> > > |LAMBADA|50.52|56.13|85.52| 89.86|
> > > |AG News|76.05|77.35|128.29 |136.73|
> > > |PubMed|51.78|50.04|74.35|55.20|
> > > |Arxiv|60.76|57.07|89.34|61.86|
> > >
> > > Throughput is measured under confidence-based parallel sampling [6], where tokens with prediction confidence ≥95% are accepted in parallel at each step.
> > >
> > > ### **2. Faster generation**
> > >
> > > Diffusion models can output multiple tokens in parallel, yielding better hardware utilization. At ~2.2× parallelism, SetDLM achieves 96.8 tok/s, a 1.8× speedup over AR (54.05 tok/s), while retaining 63.68% GSM8K accuracy. At ~2.5× parallelism, it reaches 107.2 tok/s, a 2.0× speedup over AR, at 58.23% accuracy. Distillation for discrete diffusion has been shown to enable 8× faster sampling than AR with comparable quality [7], suggesting even better tradeoffs are achievable.
> > >
> > > We hope this new evidence (Pareto dominance over BD3LM and lower perplexity than AR on 4/7 held-out benchmarks) demonstrates a meaningful contribution to the diffusion LM frontier. **Given that the speed–quality tradeoff is universal across diffusion methods and SetDLM handles it better than prior work, we respectfully ask the reviewer to consider whether these results support reconsideration of the score. Many thanks for your valuable guidance!**
> > >
> > > [1] Arriola et al "Block diffusion" 2025
> > >
> > > [2] Li et al “ReFusion” 2026
> > >
> > > [3] Sahoo et al “Esoteric Language Models” 2025
> > >
> > > [4] Sahoo et al "Simple and effective masked diffusion language models" 2024
> > >
> > > [5] Radford et al "Language Models are Unsupervised Multitask Learners" 2019
> > >
> > > [6] Wu et al "Fast-dLLM" 2025
> > >
> > > [7] Deschenaux et al "Beyond Autoregression: Fast LLMs via Self-Distillation Through Time" 2025

---

### Official Review · Reviewer_uS4K · 2026-03-13

**Soundness:** 3
**Presentation:** 2
**Significance:** 3
**Originality:** 4
**Overall Recommendation:** 4
**Confidence:** 3

**Summary:**

This paper studies how to leverage and manipulate masking orders in masked diffusion models (MDLMs) to improve both performance and efficiency. Specifically, the authors first introduce a per-token noise injection approach and, based on this mechanism, propose an order-informed noise process to impose a masking-order prior. The token ordering is further utilized to enable causal masking during training and KV caching during inference, leading to improved computational efficiency.

**Compliance With Llm Reviewing Policy:**

Affirmed.

**Final Justification:**

My previous concerns have been fully resolved.

While this approach does not yet bridge the performance gap between DLLMs and ARs thoroughly (which is hard for most DLLMs under fair comparison), its technical novelty and performance gains make a compelling case for acceptance.

**Key Questions For Authors:**

1. It seems that the KV cache implementation does not explicitly account for mask tokens. As described in Algorithm 1, $x^O$ is a subsequence from o to o+S, which should contain mask tokens. Once a mask token is unmasked, the KV of all tokens should be changed due to the bidirectional attention. Do the authors re-sort the sequence according to the decoding order and place all mask tokens at the end, and then apply the attention pattern in Fig. 8 when evaluating $x_\theta(x^O)$? If so, the paper may benefit from a more detailed explanation of this process.

2. Since the order of tokens in the sequence treated with causal attention is determined by the decoding order, how is the "ordering" of tokens in the prompt $x^{1:P}$ determined during inference? This question arises because the prompt does not undergo a decoding process. Do the authors simply apply left-to-right causal masking for the prompt, use bidirectional attention, or randomly permute the tokens in the prompt (w/o permuting position ids)?

3. A rough perspective on the training efficiency issue of MDLMs lies in the relatively low proportion of tokens that contribute to the training loss. In the proposed training strategy, 50% of the tokens in the training sequences are masked and used to compute the loss (i.e., trained tokens), while computations on the remaining clean tokens do not contribute to the optimization objective (i.e., wasted tokens). In conventional MDLM training, the expected proportion of trained tokens is also around 50%. As a reference, 100% of the tokens are trained in AR models. Could the authors further clarify how the proposed method improves training efficiency under this setting?

4. For Prop. 4.1 and its proof, could authors further clarify or highlight the equations that directly demonstrate the tightness of NELBO? In Appendix A.2, the objective is shown to be equivalent to the AR NLL when s->L. How does this result imply the tightness of NELBO?

**Limitations:**

Some of the points raised in the Key Questions and Weaknesses sections may be considered potential limitations of the work.

**Strengths And Weaknesses:**

**Strengths**

+ Incorporating an ordering prior via a windowed noise scheduler is a creative idea, and it can modulate the level of “autoregressivity” through the window size.

+ The authors leverage token ordering to achieve improved training and inference efficiency via causal attention and KV cache, respectively.

**Weaknesses**

- The paper does not present scaling experiments. The largest model evaluated in the paper appears to be 110M parameters, which is relatively small for demonstrating the capacity of modern language models.

- The presentation of the paper is relatively weak compared to the soundness of the method. It would be helpful to include illustrative figures demonstrating the training and inference processes so that readers can better understand the proposed method.

- The token ordering induced by per-token noise appears to mainly support a left-to-right decoding order prior, which may limit generalization to other ordering priors, such as anchor-guided ordering [1].

[1] Rout, Litu, Constantine Caramanis, and Sanjay Shakkottai. "Anchored diffusion language model."

---

> ### Author Rebuttal · Authors · 2026-03-31
>
> Thank you for the questions. Below we address the issues raised.
>
> # W1: Lack of Scaling Experiments
> We correct a factual point: our largest model is already at the 1.7B scale (Fig. 1, Tab. 1), not 110M. Scaling beyond 1.7B is outside our academic compute resources. SetDLM improves over diffusion baselines at both 110M and 1.7B across LM1B/OWT likelihood, infilling, summarization, GSM8K, and new commonsense evaluation (see response to reviewer zBzR W1).
>
> Consistent gains from 110M to 1.7B (15x range) together with scale-invariant properties (full KV caching, all-token training with lower gradient variance) suggest promise for further scaling. Our choice of 110M for likelihood evaluation and infilling follows MDLMs [1, 2, 5].
>
> # W2: Training/Inference Figures
> We provide new figures for training and inference (Algs 1,2) at this [anonymous link](https://docs.google.com/document/d/e/2PACX-1vQpBkXi73OW5NiZEMdTiEDDPtwoqlBkuP_j3yfjPfWXfgigsp_O-hL69WUD2Jc0Cf2AhrnHNYY7Cb_0/pub), to be added in the final version. In summary: training estimates all $p\_\theta (x^{\sigma(n)} | \mathbf{x}^{\sigma(<n)})$ for a trajectory $\sigma$ in 1 call, reducing gradient variance by ~3x; inference biases towards L2R via per-token noise schedules, unlike block diffusion [1] which must fully decode each block before advancing.
>
> # W3: Generalization to Learned Orders
> We clarify that SetDLM supports fixed and data-dependent schedules, as the objective (Eq. 3) imposes no restriction on $q(σ | \mathbf{x})$: schedules can be uniform, position-dependent, or input-dependent.
>
> Our current instantiation fixes the per-token unmasking times $a^i$ to bias toward L2R orders, motivated by the empirical success of block diffusion [1] and to avoid backpropagating through discrete sampling. This is a training choice, not a framework limitation. In principle, $a^i$ could be replaced with learned, input-dependent offsets to realize importance-based orderings (as in anchor-guided diffusion [3]) using specialized gradient estimators such as REINFORCE.
>
> Unlike anchor-guided diffusion, which applies ordering guidance at inference only and does not support KV caching or variable-length generation directly, SetDLM trains with per-token schedules that provably tighten the NELBO (Prop 4.1, App A.2) while natively supporting KV caching and variable-length generation.
>
> # Additional Questions
> > KV cache w/ mask tokens
>
> We clarify that SetDLM **does not** use bidirectional attention among query (masked) tokens (Fig. 8): each query token attends only to itself and to clean tokens preceding it in σ, independently estimating $p\_\theta(x^{\sigma(i)} | \mathbf{x}^{\sigma(<i)})$ conditioned solely on previously revealed tokens. Since there is no attention between query tokens, only the KVs of each unmasked token are added to the cache, and query tokens only attend to the cache.
>
> > Why all-token training improves efficiency
>
> The key is that "50% masked" in the input does **not** mean "50% supervised." In our Alg. 2, every original token appears once as a query target under $\sigma$, so the loss sums over all $L$ conditionals $p\_\theta(x^{\sigma(i)} | \mathbf{x}^{\sigma(<i)})$; the clean tokens are context for some queries and targets for others. MDLM instead computes cross entropy losses for the currently masked token subset. **This yields ~3× lower gradient variance (Tab. 6) and better convergence at matched training budgets (Tab. 7).**
> > Why the AR-limit result implies tightness
>
> The tightness argument has 2 parts:
>
> 1. **Schedule dependence (App A.1).** Unlike the univariate schedule in MDLM [2], SetDLM’s per-token schedule makes the NELBO path-dependent, so different schedules can yield different variational gaps.
> 2. **AR-limit equality (App A.2).** When $w = 1/L$, the unmasking windows are disjoint, only one coordinate can change at any $t$, and the posterior becomes a point mass on the next clean token. The KL term therefore reduces exactly to $-\log p\_\theta(x^j|\mathbf{x}^{<j})$, so the objective equals the AR NLL. This shows that in the AR limit the schedule-induced variational gap vanishes.
>
> Tab. 5 confirms that reducing $w$ lowers the Monte Carlo estimates of the NELBO empirically:
>
> **LM1B NELBO tightening with decreasing w (110M, 65B tokens):**
> |w|PPL (↓)|
> |-|-|
> |0.20|≤31.78|
> |0.11|≤29.91|
> |0.06|≤27.85|
> |1/L|22.83|
>
> > Prompt ordering at inference
>
> Prompt tokens are clean L2R context, cached once at decoding start (Alg. 1 line 2-3). We initialize σ with prompt positions ${1,\dots,P}$. Prompt tokens are never permuted during training/inference.
>
> **References**
>
> [1] Arriola et al. “Interpolating autoregressive and discrete denoising diffusion language models” 2025
>
> [2] Sahoo et al. “Simple and effective masked diffusion language models” 2024
>
> [3] Rout et al. “Anchored Diffusion Language Model” 2025
>
> [4] Lou et al. “Discrete Diffusion Modeling by Estimating the Ratios of the Data Distribution” 2025
>
> [5] Guo et al. “Reviving Any-Subset Autoregressive Models” 2025

---

> > ### Author Rebuttal · Reviewer_uS4K · 2026-04-04
> >
> > My concerns have been fully resolved. I will maintain my positive rating.

---

> > > ### Author Response · Authors · 2026-04-08
> > >
> > > Thank you for the positive assessment! We are glad that the additional experiments and clarifications fully addressed your concerns. We will incorporate all discussed improvements, including the added figures and implementation clarifications, in the final version.
> > >
> > > If you feel that the rebuttal and planned revisions strengthen the paper beyond your initial assessment, we would be grateful if that could be reflected in the final score. Many thanks for your thoughtful engagement and guidance!

---

### Official Review · Reviewer_iabd · 2026-03-22

**Soundness:** 3
**Presentation:** 2
**Significance:** 3
**Originality:** 3
**Overall Recommendation:** 4
**Confidence:** 2

**Summary:**

The paper introduces Set Diffusion, a language-modeling framework that aims to bridge AR decoding and masked diffusion by interpolating token generation orderings instead of committing to a fixed block structure. The main idea is to bias decoding toward left-to-right order while still allowing partially parallel, flexible generation. The method combines an order-informed noising process with a causal diffusion architecture, so that the model can retain key advantages of diffusion-style decoding while also supporting KV caching and variable-length generation more like AR models.

**Compliance With Llm Reviewing Policy:**

Affirmed.

**Final Justification:**

The rebuttal addresses my main concerns about the experiments.

**Key Questions For Authors:**

- Could the authors please clarify the primary focus of their novelty claim relative to existing work on order-aware or any-order generation?
- In Appendix I.1, it is stated that the 1.7B model was trained with a batch size of 1. Why was such a small batch size used? A batch size of 1 can create extremely noisy gradients. Did this prevent your model from fully converging? Can you provide results using a larger batch size (or gradient accumulation)?
- Your method requires tuning the unmasking width $w$ and scaling exponent $k$ to balance speed and quality. How sensitive is the model's performance to these specific parameters? Does applying this model to a new dataset or different hardware require an expensive grid search to find the optimal $w$ and $k$?
- Is a "-" missing in the Eq. (3)?
- Proposition 3.1 presents a potentially important connection between the diffusion NELBO and an any-order autoregressive objective. However, I found the proof difficult to fully follow, especially in the final step that rewrites the trajectory-wise token loss into the conditioning form of Eq. (3). In particular, Appendix G seems to reduce the ELBO to a sum over single-coordinate change events along the trajectory, but it is not fully clear to me how this implies the final any-order AR conditioning form, rather than a conditioning on the full diffusion state ($z_{t(i)}$). Could the authors please clarify this step more explicitly, ideally by stating the key intermediate lemma(s) and the precise conditioning argument? It would also be very helpful if the authors could provide a brief high-level takeaway of the proof technique itself.

**Limitations:**

yes

**Strengths And Weaknesses:**

Strength:
- The paper’s main conceptual choice is interesting: interpolating via token ordering rather than fixed block structure.
- The theoretical mapping from the continuous diffusion NELBO to the any-order AR objective via per-token schedules is interesting.

Weakness:
- Although the paper compares MDLM and Block Diffusion, its main novelty is the "interpolation-via-ordering" idea. Therefore, it would be helpful to include a more detailed comparison or discussion with other "order-agnostic" or "any-order" autoregressive models mentioned in the related work.
- See questions.

---

> ### Author Rebuttal · Authors · 2026-03-31
>
> Thank you for your feedback. We address concerns below.
> # W1: Novelty relative to any-order/order-aware work
> We add new comparisons to any-order and order-aware models, to be added to the final version. Unlike prior AOARMs [3] or order-aware models [1,2] that rely on heuristic ordering procedures, SetDLM introduces a per-token noise parameterization that interpolates between AR and fully parallel diffusion in closed form (Prop 4.1), solving three challenges:
> - **Closed-form schedule tuning.** Kim et al. [1] use random token swaps with no closed-form schedule; MAC [2] and AR Diffusion [3] cannot flexibly interpolate between ordering distributions. SetDLM tunes L2R bias in closed form
> - **KV caching.** [1, 2] cannot use KV caching due to bidirectional architecture; our causal architecture caches KV states as tokens are revealed
> - **All-token training.** [1, 2] have high variance training as they only update gradients for mask tokens; our training reduces gradient variance by ~3× (Tab 6) by predicting all $p\_\theta (x^{\sigma(n)}|\mathbf{x}^{\sigma(<n)})$ for a trajectory $\sigma$ in 1 call
> ||Tunable quality|Any-length gen.|Infilling|Full KV cache|All-token training|
> |-|-|-|-|-|-|
> |Kim et al.[1]|✓|-|✓|-|-|
> |MAC[2]|-|-|✓|-|-|
> |AR Diff.[3]|-|✓|✓|✓|✓|
> |Block Diff.[4]|✓|✓|-|-|-|
> |SetDLM (Ours)|✓|✓|✓|✓|✓|
> # W2: Proof intuition for Prop 3.1
> **High-level takeaway.** The proof re-expresses the ELBO (originally a sum over $T$ diffusion transitions) as a sum of $L$ autoregressive terms by showing that (i) only "unmasking" events contribute, (ii) as $T \to \infty$ each token is unmasked exactly once, yielding exactly $L$ events, and (iii) $\mathbf{z}_t$ at each unmasking event is informationally equivalent to the revealed set $\mathbf{x}^{\sigma(<i)}$.
>
> The intermediate lemma is: **at the unique unmasking event of token $\sigma(i)$, conditioning on diffusion state** $\mathbf{z}_t$ **is equivalent to conditioning on clean previously revealed tokens** $\mathbf{x}^{\sigma(<i)}$. This is because $\mathbf{z}_t$ is a deterministic function of $\mathbf{x}^{\sigma(<i)}$ together with [MASK] elsewhere; masked positions carry no unrevealed-token information. Hence $p\_\theta(x^{\sigma(i)}|\mathbf{z}_t) = p\_\theta(x^{\sigma(i)}|\mathbf{x}^{\sigma(<i)})$, matching Eq 3.
>
> Formally:
> 1. **Telescoping the NELBO.** We telescope the finite-T NELBO into per-transition KL ratios (Eqs. 52–60). Only unmasking events ($z_s=x,z_t=m$) contribute a nonzero term: $-\log p\_\theta(x|z_t)$. Cases $(z_s=m, z_t=m$) and $(z_s=x, z_t=x$) vanish.
> 2. **Continuous limit.** When T → ∞, each ELBO transition changes at most one dimension [6]. For the only non-trivial case $(z_s = x, z_t = m)$ the KL ratio simplifies to $-\log p\_\theta(x|z\_t)$ (Eq. 64). All other cases vanish.
> 3. **Why conditioning on $\mathbf{z}_t$ reduces to** $\mathbf{x}^{\sigma(<i)}$. At the unmasking event for token σ(i), $\mathbf{z}_t$ equals $\mathbf{x}^{\sigma(<i)}$ (clean) plus $\mathbf{m}^{\sigma(\geq i)}$ (masked). Since [MASK] carries no token-identity information, conditioning on $\mathbf{z}_t$ reduces to $\mathbf{x}^{\sigma(<i)}$, recovering Eq 3.
> # W3: GSM8K Batch Size
> The batch size follows [5], which aims to preserve the pretrained Qwen capabilities during diffusion finetuning. Our concern was catastrophic forgetting from aggressive updates, and within our compute budget larger-batch training did not improve convergence:
>
> **GSM8K Val PPL (↓) of 1.7B Models:**
> ||Batch size 1|Batch size 16|
> |-|-|-|
> |SetDLM S≤8|**1.42**|1.60|
> |SetDLM S≤32|**1.59**|1.79|
> # W4: Sensitivity of noise schedule params
> The user need only tune scalar inference budget $\bar{C}$. The decoding width $w$ should be maximized to permit any-order decoding, for which we use default $w = 4\bar{C}/L$ as the most robust setting (see below), and compute deterministically $k = \frac{\bar{C}}{Lw-\bar{C}}$ (App C2). Thus, normal use does not require a 2D search over $(w,k)$
>
> Compared to block diffusion, our default maximum lookahead (from choice of $w$) is $2S$ (2x that of block diffusion), not “25” as stated in the review.
>
> Fixing $\bar{C}$, we report GSM8K accuracy for 1.7B models with varying $w$:
>
> |w|2x parallel|3x parallel|4x parallel|
> |-|-|-|-|
> |$2\bar{C}/L$|60.65|55.04|41.24|
> |$4\bar{C}/L$|61.87|**58.38**|**49.51**|
> |$8\bar{C}/L$|**63.46**|58.23|49.28|
>
> Our default $w=4\bar{C}/L$ is most robust across the full parallelism range.
>
> # Additional Q's
> Thank you for pointing out the missing minus sign in (3); we will correct it.
>
> **References**
>
> [1] Kim et al. “Train for the worst, plan for the best” 2025
>
> [2] Shih et al. “Training and inference on any-order autoregressive models the right way” 2022
>
> [3] Hoogeboom et al. “Autoregressive Diffusion" 2022
>
> [4] Arriola et al. “Interpolating autoregressive and discrete denoising diffusion language models” 2025
>
> [5] Arriola et al. “Encoder-Decoder Diffusion Language Models” 2025
>
> [6] Campbell et al. “Continuous time framework for discrete denoising" 2022

---

> > ### Author Rebuttal · Reviewer_iabd · 2026-04-04
> >
> > Thank you for the rebuttal and additional results. I had one follow-up question: Was the batch-size-16 setting compared under the same learning rate, optimizer-step budget, and total token budget, or were these retuned as well? It would also be helpful to know whether the same conclusion holds for final GSM8K accuracy (pass@1), not just validation PPL.

---

> > > ### Author Response · Authors · 2026-04-08
> > >
> > > Thank you for the follow-up question. We clarify the controls in the original comparison and provide the requested GSM8K pass@1 results.
> > >
> > > ## **W3 follow-up: Hyperparameter sweep w/ pass@1 results**
> > >
> > > Our original batch size comparison in W3 fixed the learning rate at `1e-5` (following E2D2 [1]) across both batch sizes to isolate the effect of batch size, as our academic compute budget did not permit a full sweep at the time. Each run was evaluated at its own best-validation-PPL checkpoint, since all models eventually overfit to the GSM8K training data before the maximum number of optimizer steps: the optimizer-step and total-token budgets therefore are individually optimized per run.
> > >
> > > To test whether a larger batch helps after retuning, we provide a new sweep over learning rates {`1e-4`, `1e-5`, `1e-6`} for both batch sizes and report both validation PPL and final GSM8K pass@1. All runs start from the same Qwen-initialized 1.7B checkpoint and are again selected by best validation PPL.
> > >
> > > **GSM8K Val PPL (↓) of 1.7B Models under varying learning rates and batch sizes:**
> > >
> > > | Model / LR | | Batch size 1 | Batch size 16 |
> > > |---|---|---|---|
> > > | SetDLM S≤8, LR=1e-4 | | 1.71 | 1.55 |
> > > | SetDLM S≤8, LR=1e-5 | | **1.42** | 1.60 |
> > > | SetDLM S≤8, LR=1e-6 | | 1.51 | 1.56 |
> > > | | | | |
> > > | SetDLM S≤32, LR=1e-4 | | 1.95 | 1.70 |
> > > | SetDLM S≤32, LR=1e-5 | | **1.59** | 1.79 |
> > > | SetDLM S≤32, LR=1e-6 | | 1.81 | 1.78 |
> > >
> > > **GSM8K Pass@1 Accuracy (↑) of 1.7B Models under varying learning rates and batch sizes.** Accuracy is measured under confidence-based parallel sampling [2] for ~2x sample parallelism:
> > >
> > > | Model / LR | | Batch size 1 | Batch size 16 |
> > > |---|---|---|---|
> > > | SetDLM S≤8, LR=1e-4 | | 26.31 | 52.84 |
> > > | SetDLM S≤8, LR=1e-5 | | **64.44** | 64.14 |
> > > | SetDLM S≤8, LR=1e-6 | | 59.21 | 62.77 |
> > > | | | | |
> > > | SetDLM S≤32, LR=1e-4 | | 20.17 | 37.00 |
> > > | SetDLM S≤32, LR=1e-5 | | **58.23** | 52.08 |
> > > | SetDLM S≤32, LR=1e-6 | | 49.66 | 56.48 |
> > >
> > > **Best-checkpoint training budget for each run** (rounded optimizer steps; training tokens in M, with 1024 tokens/sequence):
> > >
> > > | Model / LR | | Batch size 1 | Batch size 16 |
> > > |---|---|---|---|
> > > | SetDLM S≤8, LR=1e-4 | | 15k steps (15.4M) | 9k steps (147.5M) |
> > > | SetDLM S≤8, LR=1e-5 | | 24k steps (24.6M) | 13k steps (212.9M) |
> > > | SetDLM S≤8, LR=1e-6 | | 43k steps (44.0M) | 21k steps (344.1M) |
> > > | | | | |
> > > | SetDLM S≤32, LR=1e-4 | | 16k steps (16.4M) | 11k steps (180.2M) |
> > > | SetDLM S≤32, LR=1e-5 | | 32k steps (32.8M) | 15k steps (245.8M) |
> > > | SetDLM S≤32, LR=1e-6 | | 49k steps (50.2M) | 27k steps (442.4M) |
> > >
> > > These results suggest that after LR retuning, batch size 16 does not improve over our batch size 1 setting. Comparing each batch size at its best learning rate: for S≤8, the best pass@1 scores are comparable with batch size 16 requiring ~8.7× more training tokens (64.44 at BS1/LR=`1e-5` vs. 64.14 at BS16/LR=`1e-5`), while for S≤32, batch size 1 remains better (58.23 at BS1/LR=`1e-5` vs. 56.48 at BS16/LR=`1e-6`). The same pattern holds for validation PPL.
> > >
> > > We note that for BS16 at LR=`1e-4`, the best-val-PPL checkpoints achieve poor pass@1 for both S≤8 (52.84) and S≤32 (37.00), suggesting that the high learning rate combined with the large effective batch leads to catastrophic forgetting of the base Qwen model's reasoning capabilities.
> > >
> > > The best batch-size-16 checkpoints also require substantially more training tokens despite fewer optimizer steps (e.g., for S≤8 at LR=`1e-5`: 13k vs. 24k steps, but 212.9M vs. 24.6M tokens). We therefore do not see evidence in this 1.7B GSM8K finetuning setting that increasing the batch size from 1 to 16 improves final accuracy or convergence quality.
> > >
> > > ---
> > >
> > > We hope this fully resolves the remaining W3 question. Together with our initial rebuttal addressing W1, W2, W4, we believe all raised concerns have now been addressed. We thank the reviewer for prompting us to add the pass@1 and LR-retuning controls, which strengthen the paper. **We respectfully ask the reviewer to consider whether these results support reconsideration of the score. Many thanks for your valuable guidance and constructive engagement throughout!**
> > >
> > > [1] Arriola et al. “Encoder-Decoder Diffusion Language Models” NeurIPS 2025
> > >
> > > [2] Wu et al. "Fast-dLLM: Training-free Acceleration of Diffusion LLM by Enabling KV Cache and Parallel Decoding" ICLR 2026.

---

### Decision · Program_Chairs · 2026-04-30

**Decision:**

Accept (regular)

**Comment:**

This paper proposes Set Diffusion, a diffusion language modeling framework that replaces fixed block-based decoding with a stochastic  order generation process that interpolates between autoregressive and fully random denoising. The method combines an order-informed noise process with a causal diffusion architecture that enables KV caching under non-deterministic token orderings, allowing decoding in sliding-window token sets. Empirically, the approach is intended to improve the speed-quality tradeoff of diffusion LMs while retaining greater flexibility for infilling and any-order generation than prior block-based approaches.

Reviewers generally found the paper’s core conceptual idea compelling. The most positively received aspect is the decision to interpolate between autoregressive and diffusion-style generation through token ordering, rather than through a fixed block structure. Moreover, the theoretical framing of this paper is greatly appreciated.

The most common weakness is around two perspectives. On one hand, the novelty relative to prior work is questioned, and reviewers wanted a more careful comparison and discussion of related approaches, especially those that also model or learn decoding order. A second major concern is the limited experimental scope, especially regarding scale. Several reviewers noted that much of the evaluation is on relatively small models, and even where the paper includes a 1.7B model, the evaluation appears in narrow scope. Reviewers requested broader downstream results at larger scales, as well as clearer evidence that the method remains competitive as model size increases.

The final reviewer distribution was two weak accepts and two weak rejects, with no strong accept or strong reject. Reviewers broadly agreed that the method is novel and technically sound. The main concern was the scale of the experiments and the still substantial performance gap to strong autoregressive baselines, rather than any major flaw in the method itself.

In the final author comments, the authors provided more comprehensive results on1.7B-scale experiments, which materially strengthened the empirical case. Based on these additional results, the AC’s view is that the paper is solid and justified at least at the 1.7B scale. While evaluation at even larger scales would still be desirable, the current results are sufficient to validate the correctness and promise of the proposed method. Although bigger scales models are not tested at this moment, it is good enough to demonstrate the correctness of the proposed method. It would be encouraged to add results across different scales in the camera-ready if possible.